# Prototype-Grounded Concept Models for Verifiable Concept Alignment

Stefano Colamonaco [* 1]   David Debot [* 1]   Pietro Barbiero [2]   Giuseppe Marra [1]

## Abstract

Concept Bottleneck Models (CBMs) aim to improve interpretability in Deep Learning by structuring predictions through human-understandable concepts, but they provide no way to verify whether learned concepts align with the human's intended meaning, hurting interpretability. We introduce Prototype-Grounded Concept Models (PGCMs), which ground concepts in learned visual prototypes: image parts that serve as explicit evidence for the concepts. This grounding enables direct inspection of concept semantics and supports targeted human intervention at the prototype level to correct misalignments. Empirically, PGCMs achieve similar predictive performance as state-of-the-art CBMs while substantially improving transparency, interpretability, and intervenability.

## 1. Introduction

Modern neural networks achieve remarkable predictive performance, yet their lack of semantic transparency remains a major obstacle to trustworthy deployment. While many Explainable AI methods aim to explain individual predictions post-hoc, an increasingly influential line of work instead builds interpretable models (Espinosa Zarlenga et al., 2022; Mahinpei et al., 2021; Barbiero et al., 2023; Vandenhirtz et al., 2024; Yuksekgonul et al., 2022). In this paradigm, predictions are structured through human-understandable intermediate representations, also called *concepts*, which enables human-AI interaction in the form of inspection, verification, and intervention. Arguably the most well-known of these are Concept Bottleneck Models (CBMs) (Koh et al., 2020). In CBMs, an input is first mapped to a set of high-level, symbolic concepts (e.g. *has stripes*, *is metallic*), and

the final prediction is computed exclusively from these concept predictions, typically via a simple and transparent classifier. This design offers strong interpretability guarantees: users can inspect concept activations, intervene by modifying them, and reason about predictions entirely at the level of high-level concepts.

Despite these appealing properties, CBMs suffer from a fundamental limitation: their concepts lack low-level grounding. Even when concepts are directly supervised using human-provided labels, there is no guarantee that the learned representation aligns with the intended semantics. A concept labeled *has stripes* may in fact rely on spurious textures, background cues, or correlated artifacts. Crucially, users have no direct way to verify this alignment, because the visual or low-level evidence underlying a concept prediction remains hidden. Post-hoc explanation methods such as saliency maps do not allow this either, as they only explain individual predictions rather than providing interpretability for the entire model. As a result, CBMs are only interpretable under a strong and often unjustified assumption of concept alignment: that the learned concepts mean what their names suggest.

In this work, we address this limitation by explicitly grounding concepts in concrete visual evidence. We introduce Prototype-Grounded Concept Models (PGCMs), a framework that augments Concept Bottleneck Models with learned visual prototypes. In a PGCM, each concept is not represented solely as an abstract scalar prediction, but is instead associated with a set of learned prototypes: localized visual patterns that serve as concrete exemplars of what the model considers evidence for that concept. At inference time, a PGCM explains its concept predictions in terms of similarity to these prototypes. Concept activations are thus derived from and can be inspected through the visual exemplars most relevant to the input. As a result, each concept is given a dual representation: a high-level symbolic label (e.g. *grey hair*) and a collection of low-level visual instances that make this meaning explicit (e.g. specific image instances of *grey hair*). This integration embeds visual grounding directly into the concept bottleneck.

This grounding yields several important advantages. First, PGCMs enable verification of concept alignment: users can directly examine the prototypes associated with each

---

[*]Equal contribution  [1]Department of Computer Science, KU Leuven [2]IBM Research, Zurich. Correspondence to: Stefano Colamonaco <stefano.colamonaco@kuleuven.be>, David Debot <david.debot@kuleuven.be>.

*Proceedings of the 43$^{rd}$ International Conference on Machine Learning*, Seoul, South Korea. PMLR 306, 2026. Copyright 2026 by the author(s).

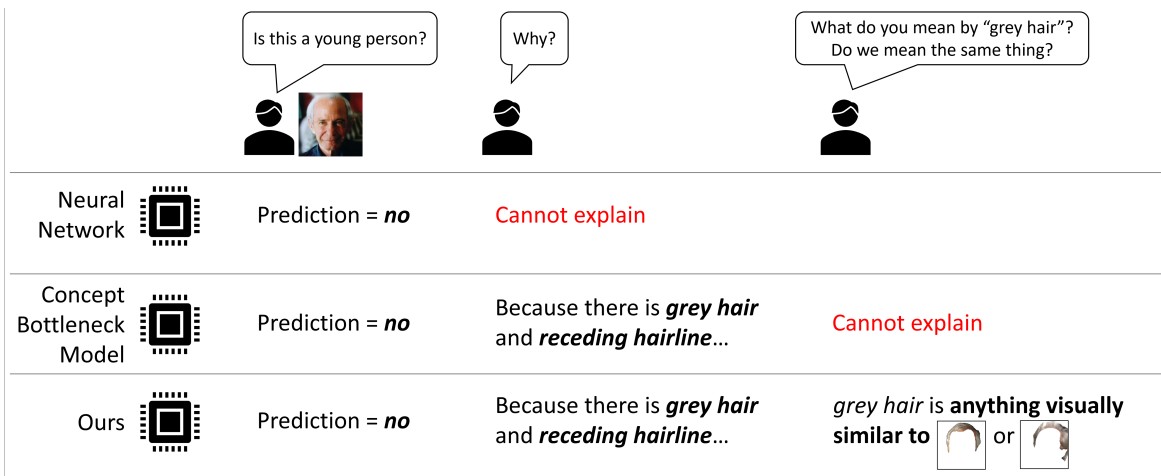

*Figure 1.* Comparison between standard neural networks, Concept Bottleneck Models (CBMs), and our Prototype-Grounded Concept Models (PGCMs). While CBMs explain predictions in terms of high-level concepts, they provide no way to verify what those concepts mean visually. PGCMs ground each concept in learned visual prototypes, making concept semantics explicit and inspectable.

concept to assess whether the learned semantics match their intended semantics. Second, the dual representation of concepts unlocks new forms of human intervention. Rather than intervening only on individual concepts, as in standard CBMs, users can intervene at the prototype level (removing, modifying, or selecting prototypes) which can simultaneously correct multiple concept predictions. Third, PGCMs expand the explanatory power of CBMs, enabling explanations that reference both abstract concepts and concrete visual evidence. Our code is available at https://github.com/daviddebot/PGCM.

*In summary, while Concept Bottleneck Models promise interpretability through abstraction, they ultimately require trust in unobservable semantics; PGCMs replace this assumption with verifiable, visually grounded concepts (Figure 1).*

## 2. Background

Concept Bottleneck Models (CBMs) are deep learning models that provide interpretability by predicting the target in two steps. First, they map the low-level input features (e.g. pixels) to a set of high-level *concepts* (e.g. *red*, *dog*) using a neural network. Second, they map the predicted concepts on the target task using either an interpretable function, e.g. a linear layer, or a black-box neural network. The concepts form a high-level, semantically meaningful abstraction through which the model can be interpreted or explained. To align concepts to the meaning intended by the human, they are typically explicitly supervised during training. A core advantage of CBMs is the ability to do *concept interventions*: human experts may at test time correct mispredicted concepts, which may influence the downstream task prediction, making it more accurate.

**Notation.** We write random variables in uppercase (e.g. $p(C)$) and their assignments in lower case (e.g. $p(C = c)$). When clear from the context, we abbreviate assignments (e.g. $p(C = c)$ becomes $p(c)$). We use curly brackets to write groups of random variables concisely (e.g. $p(\{C_i\}_{i=1}^3) = p(C_1, C_2, C_3)$). We follow standard variational inference notation, with $p(.)$ denoting generative distributions and $q(.)$ denoting variational posteriors.

## 3. Model

We propose Prototype-Grounded Concept Models (PGCMs), a hybrid architecture that predicts high-level concepts by grounding the prediction in low-level prototypes. PGCMs retain the familiar CBM structure (input $\rightarrow$ concepts $\rightarrow$ task) while innovating in how concepts are inferred from input data. As is typical in CBMs, the concepts $C$ are given by the human and directly supervised, i.e. the dataset not only has task labels, but also concept labels. In practice, the need for concept supervision is not a significant limitation as it can come from Vision-Language Models instead of being provided by humans (Oikarinen et al., 2023).

### 3.1. Overview

In this section, we provide a high-level description of the model. The parametrization of the different distributions are given in Section 3.2, and additional details (e.g. the neural architectures) are given in Appendix A.1. At a conceptual level, the model can be viewed as a three-stage mapping:

$$x \rightarrow p \rightarrow c \rightarrow y,$$

where $x$ is the input image, $p$ is a set of similarity scores to learned prototypes, $c$ is the concept representation of $x$, and $y$ is the task prediction.

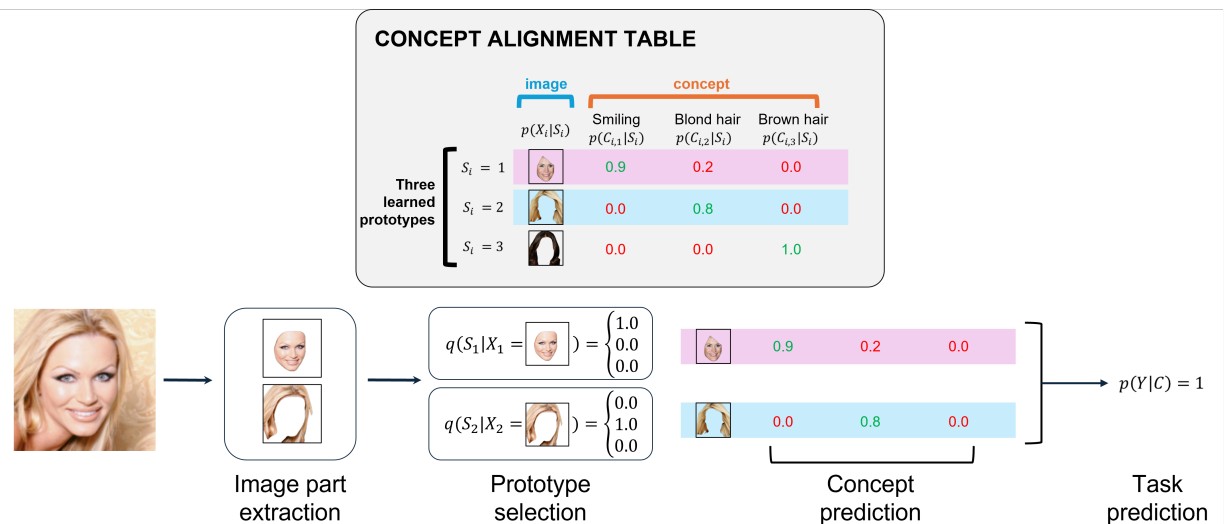

*Figure 2.* Example inference of PGCMs. From the image $X$, we use a segmentation model to extract two important image parts $X_1 =$ 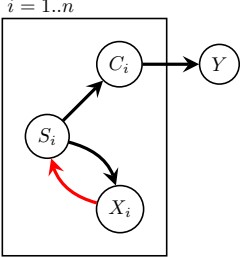
and $X_2 =$ 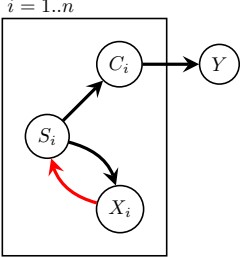. For each image part, the *prototype selector* $q(S_i|X_i)$ predicts a categorical distribution over the three learned prototypes.
For instance, for 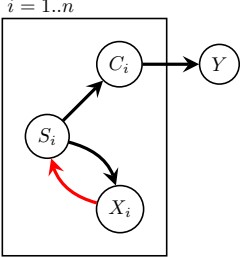, it assigns all probability to the first prototype. Our *Concept Alignment Table* shows the prototypes and their different
representations: an image representation defined by the *image decoder* $p(X_i|S_i)$ and a concept representation defined by the *concept
decoder* $p(C_i|S_i)$. For each image part, we pick the concepts of the selected prototype, which together form our predicted concepts. The
task predictor $p(Y|C)$ then maps all predicted concepts on the target task.

As a preprocessing step, we extract *image parts* from each
input image using a segmentation network. Concretely,
each image $x$ is mapped to $n$ image parts $\{x_i\}_{i=1}^n$, with $n$ a
hyperparameter. The motivation is to isolate spatial regions
that are informative for predicting high-level concepts. We
view segmentation similarly to concept supervision: an
additional effort that is justified by the goal of achieving
concept alignment, which is our primary concern.

The probabilistic graphical model (PGM) of our model is
shown in Figure 3. Our model combines generative inference (black arrows) and discriminative inference (red and
black arrows), which serve complementary purposes.

The **generative model** is primarily used to inspect and interpret the learned prototypes. For each image part $i$, we
introduce a latent variable $S_i$, representing the *selection*
of a prototype. Its prior $p(S_i)$ is a categorical distribution
with one value per learned prototype. The **image decoder**
$p(X_i|S_i)$ maps a prototype index $S_i = j$ to a concrete
image part, which we call the *image representation* of prototype $j$. In parallel, the **concept decoder** $p(C_i|S_i)$ maps
the prototype index to a distribution over high-level concepts, yielding the prototype's *concept representation*. By
enumerating all possible values of $S_i$, we can visualize each
prototype via the image decoder, and inspect its concepts via
the concept decoder. This provides the underlying formal
semantics of our model.

The **discriminative model** is used to make concept and task
predictions given an input image $x$. In this setting, the image

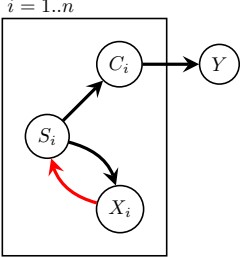

*Figure 3.* Probabilistic Graphical Model (PGM) of PGCMs. Black
arrows are used for generative and discriminative inference. Red
arrows are only used for discriminative inference.

parts $X_i$ are observed, as they are extracted deterministically from $x$ using our segmenter. The **prototype selector**
$q(S_i|X_i)$ assigns a distribution over prototypes to each image part $X_i$. Intuitively, this module selects the prototype
that is the most visually similar to the given image part.
Given a selected prototype, the **concept decoder** $p(C_i|S_i)$
maps the selected prototype on high-level concepts. Finally,
the **task predictor** $p(Y|\{C_i\}_{i=1}^n)$ is standard, and can be
e.g. a linear layer or neural network mapping the concept
predictions on the final task.

Figure 2 shows an example of inference using our PGCMs.
The segmentation model extracts two image parts 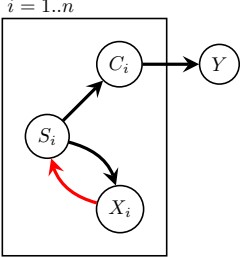 and 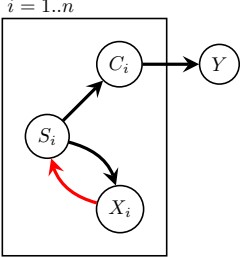.
For each part, the prototype selector assigns high probability
to the most visually similar learned prototype, in this case
assigning all probability mass to a single prototype: e.g. for
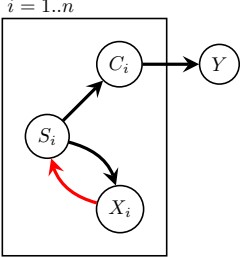, it selects the prototype with image representation 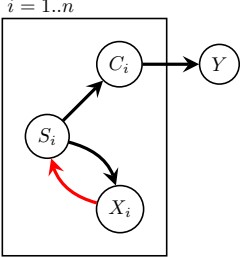.

The concept predictions for each part are then inherited from the selected prototype's concept representation. Finally, the task predictor maps the predicted concepts across parts to the final task output.

## 3.2. Parametrizations

In this section, we describe the parametrization of the distributions introduced in Section 3.1.

**Prototype embeddings.** We denote with $m$ the number of prototypes we want to learn. Each prototype $j \in \{1, \dots, m\}$ is associated with a learnable embedding $e_j \in \mathbb{R}^d$ which represents the latent embedding of prototype $j$. We use these embeddings in the parametrizations of the components below when we condition on the prototype selection $S_i$.

**Image decoder.** The image decoder models the distribution of an image part given a prototype assignment, which we model using a multivariate Gaussian with a fixed diagonal covariance matrix:

$$p(x_i \mid S_i = j) = \mathcal{N}\big(x_i \mid \mu_j, \sigma^2 I\big),$$

with $\mu_j \in \mathbb{R}^D = f_{\text{image}}(e_j)$, $f_{\text{image}}$ a neural network operating on prototype embeddings, $D$ the number of pixels in an input image, and $\sigma^2$ a fixed constant variance. This decoder defines the image representation of each prototype.

**Concept decoder.** The concept decoder maps a prototype to a distribution over high-level concepts, where each concept is modelled independently given a selected prototype:

$$p(c_i \mid S_i = j) = \prod_{l=1}^{k} p(c_{i,l} \mid S_i = j) \qquad (1)$$

$$\text{where} \quad p(c_{i,l} \mid S_i = j) = \text{Bernoulli}(\pi_{j,l})$$

with $\pi_j \in [0,1]^k = f_{\text{concept}}(e_j)$ the concepts' probabilities, $f_{\text{concept}}$ a neural network operating on prototype embeddings, and $k$ the number of concepts. This distribution defines the concept representation of each prototype. Intuitively, it performs concept classification of prototype embeddings.

**Prototype selector.** The variational posterior over prototype assignments is implemented by a prototype selector $q(S_i \mid X_i)$ where each image part $x_i$ is first deterministically mapped to an embedding $z_i = f_{\text{enc}}(x_i) \in \mathbb{R}^d$, with $f_{\text{enc}}$ a neural network. Prototype assignment probabilities are computed using a similarity measure $\text{sim}(\cdot, \cdot)$ between every prototype embedding and the image part's embedding, combined with a softmax to obtain a categorical distribution:

$$q(S_i = j \mid X_i) = \frac{\exp\big(\text{sim}(z_i, e_j)\big)}{\sum_{k=1}^{m} \exp\big(\text{sim}(z_i, e_k)\big)},$$

Intuitively, this corresponds to a prototype classification of image parts.

**Segmentation model.** We extract the image parts $\{x_i\}_{i=1}^n$ using a segmentation network that isolates (potentially overlapping) spatial regions from the input $x$. This component is flexible, as it is possible to use ground-truth masks to train it or use a pretrained segmenter in their absence.

## 3.3. Inference

At inference time, as standard in CBMs, we observe the input image $x$ and want to predict concepts and tasks. We first apply the trained segmentation model to obtain $n$ image parts $\{x_i\}_{i=1}^n$. For each image part $x_i$, we compute the posterior predictive distribution over concepts as

$$p(c_{i,j}|x_i) = \mathbb{E}_{q(s_i|x_i)}\Big[p(c_{i,j} \mid s_i)\Big] \qquad (2)$$

Each concept prediction is then $\hat{c}_{i,j} = \arg\max_{k \in \{0,1\}} p(C_{i,j} = k|x_i)$. Given the predicted concepts, the task prediction is computed as $\hat{y} = \arg\max_{y \in \mathcal{Y}} p(y|\{\hat{c}_i\}_i)$ with $\mathcal{Y}$ the set of possible classes. This means we employ *hard concepts*, avoiding the problem of concept leakage in CBMs which may occur when using *soft concepts* (Marconato et al., 2022; Havasi et al., 2022).

**Interpretability optimizations.** We want to have the hard guarantee that each prototype only encodes the information present in its image representation, and that each learned image representation is a realistic image to the human. We guarantee the former by mapping each prototype embedding to its image representation and then map the resulting prototype image back to an embedding, i.e. $e'_j = f_{enc}(f_{image}(e_j))$ (see Section 3.2 for $f_{enc}$ and $f_{image}$). Instead of using $e_j$, we use $e'_j$ both during training and inference. *This ensures that the prototype embedding cannot contain any information not present in the prototype image.* We guarantee the latter by performing a prototype swapping step halfway through training, where we replace each prototype with the closest image part in the training set. *This ensures each prototype image is a realistic image.* For more details, we refer to Appendix A.

## 3.4. Training

We maximize the joint log-likelihood of the observed image parts, concepts, and task label. Under our probabilistic graphical model, this likelihood admits the following evidence lower bound (ELBO):

$$\log p(\{x_i\}_i, \{c_i\}_i, y) \geq \qquad (3)$$
$$- \sum_i \underbrace{\text{KL}\big(q(s_i|x_i) \,\|\, p(s_i)\big)}_{\text{regularization term}} + \underbrace{\log p(y \mid \{c_i\}_i)}_{\text{task loss}}$$
$$+ \sum_i \Big( \mathbb{E}_{q(s_i|x_i)}\Big[ \underbrace{\log p(c_i \mid s_i)}_{\text{concept loss}} + \underbrace{\log p(x_i \mid s_i)}_{\text{reconstruction loss}} \big) \Big]$$

For the derivation, we refer to Appendix C. This objective decomposes into four components, two of which are the standard task and concept loss of CBMs. The concept and task components further factorize independently to each individual concept and task, e.g. $p(c_i|s_i) = \prod_{j=1}^{k} p(c_{i,j}|s_i)$.

Due to the reconstruction loss, (1) the prototype selector $q(s_i|x_i)$ is encouraged to select prototypes that are visually similar to the image part $x_i$, and (2) the learned prototypes are encouraged to be visually representative of the data. Ignoring additive constants, the reconstruction log-likelihood simplifies to $\log p(x_i|s_i) \propto -||x_i - \mu_i||^2$.

Due to the concept loss, the prototype selector is also encouraged to maximize a correct prediction of the concepts. This entails that the selector should not only select based on unsupervised, visual cues, but also using supervised, semantic cues. This is different from e.g. standard prototype-based networks, where only unsupervised visual cues and weakly-supervised task-based cues are used (see Section 5).

We use the regularization term between the posterior and the prior to encourage higher-entropy prototype selection $q(s_i|x_i)$, promoting exploration during training and avoiding local minima. We achieve this by choosing each $p(s_i)$ to be a fixed uniform categorical distribution over prototypes.

The segmentation model used to extract image parts $x_i$ from the input image $x$ can either be pretrained or learned using direct supervision on the image parts (using a standard cross-entropy loss).

## 4. Discussion

### 4.1. Semantics

Standard CBMs' concept prediction is black-box, which makes it impossible for the human to check what the meaning, i.e. *semantics*, is of the learned concepts. PGCMs resolve this by grounding the concepts in low-level, visual prototypes. We define the meaning of a concept explicitly through a **concept alignment table** (Figure 2). This table maps every prototype index $j$ to a dual representation:

- An **image representation**, generated by the image decoder $p(X_i|S_i = j)$.[1]

- A **concept representation**, generated by the concept decoder $p(C_i|S_i = j)$.

This design mirrors a classical notion of semantics from logic (Tarski, 1944), where meaning is defined by an explicit correspondence between symbols and elements of a domain. In our case, concepts play the role of the abstract linguistic symbols, prototypes form the set of concrete entities

---

[1]Specifically, we consider the mode of the Gaussian the prototype's image representation.

they refer to, and the concept alignment table specifies this correspondence. As a result, the semantics of the learned concepts is no longer implicit or purely operational, as in CBMs, but given directly by how each concept is grounded in identifiable prototypes and their visual realizations. This stands in contrast to standard CBMs, where concepts are learned as latent variables without a transparent link to what they denote, making their meaning difficult to inspect or verify. By making this correspondence explicit, PGCMs provide a clear and auditable notion of semantics for the concept space.

### 4.2. Interpretability and Alignment

**Alignment.** The human can inspect the concept alignment table to verify whether the learned concepts are aligned with their intended interpretation: for each row, they can compare the visual representation of a prototype with its associated concept representation and assess whether the listed concepts are indeed present in the image. This form of inspection is not possible in standard CBMs, where concept prediction remains a black-box process.

The low-level semantics of a high-level concept is explicitly defined as the disjunction of the image representations of all prototypes for which that concept is active. Specifically, a concept is only true *if there exists a part of the image that is similar enough to at least one of these prototypes*. For instance, if the concept alignment table indicates that only prototypes having image representations $x_1$ and $x_2$ possess the concept $C_{\text{red}}$, then the model explicitly defines the semantics of "red" as:

$$C_{\text{red}} \text{ is active} \iff (\text{looks like } x_1) \lor (\text{looks like } x_2) \quad (4)$$

**Visual explanations.** Typical CBMs provide explanations and interpretability in terms of the concepts. PGCMs can also provide this in terms of the low-level input features. In addition to showing the human which concepts led to a concrete task prediction, PGCMs show which *prototypes* led to that prediction. This is akin to typical prototype-based networks (see Section 5), which explain the output of the model in terms of prototypes, but lack high-level semantics (concepts).

**Guarantees.** In PGCMs, for a concrete prediction, parts are extracted from the image by masking the rest of the image. This gives the human the guarantee that the masked input features were not used for that prediction.

### 4.3. Intervenability

Our proposed model allows for new types of human intervention and improves existing ones. Interventions in PGCMs operate at three levels: concept labels, prototypes, and prototype selection.

**Alignment interventions.** The first novel intervention type is a specific type of *model editing* (Mazzia et al., 2024), which the human can do to re-align misaligned concepts. There are three types of such interventions possible. First, the human may *edit learned prototypes*. If the human disagrees with the concept representation of a prototype, they may *relabel* the concepts of the prototype to correct it. This relabeling simply corresponds to changing values in the table. Second, the human may *remove learned prototypes*. This is useful if the human detects spurious prototypes (using the prototypes' image representations). Finally, the human may *add entirely new prototypes* by adding one or more image representations to the set of prototypes, e.g. image parts taken from the dataset or manually crafted. $f_{enc}$ and $f_{concept}$ will map these to an embedding and concept representation.

**Improved concept interventions.** In contrast to most CBMs, concepts are not independent of each other, as they are connected through the prototypes. Therefore, a single concept intervention may correct multiple concepts at once: it can be used to set all similarity scores to zero of prototypes where that concept prediction is wrong. This may change not only the prediction for the intervened concept, but also the prediction for the other concepts, similar to some existing concept predictors with dependent concepts (Dominici et al., 2024; Vandenhirtz et al., 2024).

**Prototype interventions.** Instead of intervening on a concept prediction at test time, the human may prefer to intervene on the low-level representations, and *intervene on the prototype selection*. A first option is to assign all probability to a single prototype ("this part is actually the most similar to that prototype"). A second option is to remove all probability of one or more prototypes ("this prototype is not similar at all").

## 5. Related Work

Many extensions to Concept Bottleneck Models (CBMs) have been developed, improving their accuracy by alleviating the information bottleneck (Espinosa Zarlenga et al., 2022; Mahinpei et al., 2021; Sawada & Nakamura, 2022; Barbiero et al., 2023; Debot et al., 2024), intervenability by modeling inter-concept relations (Vandenhirtz et al., 2024; Dominici et al., 2024), and applicability by replacing concept supervision with pretrained foundation models (Oikarinen et al., 2023). However, a critical limitation that all CBMs still share is the lack of a way to inspect whether the learned concepts are correctly aligned to the human's interpretation. This is a core limitation, as the entire claim of CBMs being interpretable rests on this alignment.

Our prototype learning brings us close to *part-prototype networks*, which offer interpretability by grounding task predictions in prototypes. Prototype-based networks typically learn a set of prototypes and classify new inputs based on their similarity to these prototypes (Snell et al., 2017; Andolfi & Giunchiglia, 2025). Part-prototype networks typically identify prototypical parts within feature maps, enabling "this looks like that" explanations (Chen et al., 2019; Donnelly et al., 2022; Ma et al., 2023). However, unlike CBMs, prototype-networks lack high-level, controllable semantics. Since their prototypes are not supervised using concepts, they may capture recurrent visual patterns rather than human-aligned concepts. Moreover, part-prototype networks do not learn concrete image representations of prototypes, instead entirely relying on latent embedding spaces. Some prototype networks exist that *do* learn image representations of prototypes, but they learn prototypes of *entire images* as opposed to parts, and they do not employ concepts (Li et al., 2018). Other works attempts to ensure concept trustworthiness by grounding prototypes spatially within the input image (Huang et al., 2024; Jeon et al., 2025; Wang et al., 2024). Yet, similar to part-prototype networks, they do not provide explicit image representations of prototypes, which makes it impossible to verify concept alignment. Finally, some approaches combine prototype learning with concept discovery, and do not operate on a predefined, controlled concept set (Knab et al., 2024; Prasse et al., 2024).

The motivation to decompose visual scenes into modular components shares philosophical roots with object-centric learning (Burgess et al., 2019; Locatello et al., 2020). By explicitly isolating object-like entities, these approaches avoid entangled global representations and facilitate robust compositional reasoning (Steinmann et al., 2025; Colamonaco et al., 2025). This structural separation is a natural fit for interpretability, as it ensures that concepts can be explicitly grounded in localized, verifiable visual evidence.

## 6. Experiments

Our experiments aim to answer the following research questions: **(Interpretability and alignment)** Can humans verify whether learned concepts align with their intended semantics by examining the learned concept alignment tables? Can we intervene on this alignment? **(Accuracy)** Do PGCMs achieve similar concept and task accuracy as state-of-the-art CBMs? **(Capacity)** How sensitive are PGCMs to the number of prototypes used? **(Versatility)** Can PGCMs be used in different scenarios and with different forms of data available? **(Intervenability)** How effective are concept interventions in PGCMs?

### 6.1. Experimental Setup

This section provides essential info about experiments. For more details, we refer to Appendix B.

*Table 1.* Example rows from the Concept Alignment Table of CelebA, ColorMNIST+ and CLEVR-Hans (seed 1).

| Image | Active Concepts |
| --- | --- |
|  | Bangs, Blond Hair, Wavy Hair |
|  | Heavy Makeup, No Beard, Pale Skin |
|  | Digit 8 |
|  | Digit is 4 |
|  | Small, Cyan, Sphere, Metal |
|  | Small, Yellow, Cube, Rubber |

*Table 2.* Example concept semantics expressed as a disjunction over prototypes for CelebA, ColorMNIST+ and CLEVR-Hans (seed 1).

| Concept | Semantics |
| --- | --- |
| Brown Hair |  |
| Pointy Nose |  |
| Digit is 3 |  |
| Blue |  |
| Cube |  |

**Datasets.** We used four standard datasets for evaluating CBMs: CelebAMask-HQ (Lee et al., 2020; Liu et al., 2015), where the each input is the face of a celebrity and where the concepts are face attributes (e.g. *blond hair* or *beard*); CUB, where each input is an image of a bird, the concepts are bird features and the task is the bird class; ColorMNIST+, where each input is a pair of colored MNIST digits, and where concepts are the digits represented in the images, and the task is their sum (adapted from MNIST+ (Manhaeve et al., 2018)). We also used the CLEVR-Hans3 dataset (Stammer et al., 2021), where each input is an image consisting of different objects, where the concepts are the shape, color, size and material. Moreover, we considered a version of ColorMNIST+ and CLEVR-Hans3 where the concept labels are noisy. Concept prediction is visually hard in CelebA and CUB but easy in ColorMNIST+ and CLEVR-Hans3, representing two common settings for CBMs. These datasets represent a diverse range of settings, particularly regarding the availability and type of visual grounding. For ColorMNIST+ and CelebA, we utilize provided segmentation masks and train a segmenter. In contrast, for CLEVR-Hans3, where masks are not available, we use the Segment Anything Model (SAM) (Kirillov et al., 2023) to extract objects. Finally, in CUB, we avoid segmentation entirely by treating the full image as a single part.

**Baselines and evaluation.** We compare with Concept Bottleneck Models (CBM) (Koh et al., 2020), Concept Residual Models (CRM) (Mahinpei et al., 2021), and Concept-based Memory Reasoner (CMR) (Debot et al., 2024). In all three models, the concept predictor is a neural network. For CBM, we take a neural network as task predictor. CRM and CMR are models with a sidechannel, i.e. the task is predicted not only using concepts, but also using additional informa-

tion, which hurts interpretability (Debot & Marra, 2025). In CRM, this is an embedding predicted from the input, and the task predictor is a neural network. In CMR, this is a rule selection mechanism, and the task predictor is a memory of learned logic rules. We additionally train a deep neural network (DNN) to predict the task directly from the input as reference. We evaluate concept prediction using balanced accuracy and task prediction using regular accuracy. When we show images from CelebA, we make the black pixels transparent for visualisation purposes, and we upscale some of them.

### 6.2. Key findings

**PGCMs make concept alignment and semantics directly inspectable (Tables 1 and 2).** This is PGCMs' core advantage over existing CBMs, and can be done by inspecting the learned concept alignment table, which instantiates the formal semantics defined in Section 4.2. Table 1 shows some examples rows of the learned tables; the concepts can be clearly recognized in the image representations of the prototypes. Table 2 gives the alternative view: it shows the semantics for individual concepts as explained in Section 4.2, showing that e.g. the hairstyle does not matter for predicting brown hair, and the color and position of the digit do not matter for predicting a digit 3.

**PGCMs achieve similar levels of accuracy as standard CBMs, despite their interpretability advantages (Table 3).** By predicting concepts via a finite set of learned prototypes, PGCMs constrain model capacity compared to existing concept-based models. This may result in a gap in accuracy w.r.t. deep neural networks, and is well-known for prototype-based networks (Chen et al., 2019). In spite of

*Table 3.* Concept and task accuracy on CelebA, ColorMNIST+ and CUB over three seeds.

| Model | ColorMNIST+ | | CelebA | | CUB | |
|---|---|---|---|---|---|---|
| | **Concept** | **Task** | **Concept** | **Task** | **Concept** | **Task** |
| DNN | – | $99.2 \pm 0.1$ | – | $84.7 \pm 0.2$ | – | $64.5 \pm 0.4$ |
| CBM (Koh et al., 2020) | $99.2 \pm 0.1$ | $99.6 \pm 0.1$ | $81.3 \pm 0.4$ | $84.0 \pm 0.3$ | $89.9 \pm 0.2$ | $63.6 \pm 0.9$ |
| CRM (Mahinpei et al., 2021) | $99.0 \pm 0.0$ | $99.4 \pm 0.1$ | $76.8 \pm 0.3$ | $84.8 \pm 0.2$ | $87.1 \pm 0.3$ | $64.9 \pm 0.3$ |
| CMR (Debot et al., 2024) | $99.0 \pm 0.0$ | $99.2 \pm 0.2$ | $76.3 \pm 0.1$ | $84.7 \pm 0.5$ | $88.2 \pm 0.2$ | $63.0 \pm 0.6$ |
| **PGCM (ours)** | $99.2 \pm 0.0$ | $99.7 \pm 0.0$ | $78.5 \pm 0.1$ | $83.0 \pm 0.0$ | $90.0 \pm 0.1$ | $63.4 \pm 0.9$ |

*Table 4.* Model editing interventions (MEI): removing or editing wrong prototypes after training on noisy labels improves concept accuracy on the non-noisy test set.

| MEI | ColorMNIST+ | | CLEVR-Hans3 | |
|---|---|---|---|---|
| | Before | After | Before | After |
| Remove | $92.9 \pm 0.3$ | $96.9 \pm 0.9$ | $81.6 \pm 0.4$ | $95.1 \pm 3.1$ |
| Edit | $92.8 \pm 0.2$ | $97.8 \pm 1.4$ | $81.6 \pm 0.4$ | $97.5 \pm 1.0$ |

this, our concept and task accuracy is similar to existing concept-based models on ColorMNIST+ and CUB, with only a small drop on CelebA. We attribute the drop in task accuracy for PGCM on CelebA due to its drop in concept accuracy. CMR and CRM do not suffer a drop in task accuracy (despite having lower concept accuracy than CBM) because they have a sidechannel (i.e. they predict the task not only using concepts).

**PGCMs allow for novel model editing interventions, increasing accuracy and improving alignment (Table 4).** We evaluate PGCMs before and after performing model editing, considering two interventions: *removing* wrong prototypes and *editing* (correcting) their concepts. For these experiments, we gave ColorMNIST+'s and CLEVR-Hans3's training and validation sets noisy concept labels (see Appendix B for details). As a result, some of the learned prototypes are misaligned to the human, e.g. learning a prototype with image representation ⬛ and concept representation $c_5 = 1$, as opposed to $c_1 = 1$. By dropping the wrong prototypes or editing their concept representation, accuracy on the non-noisy test set significantly increases. *Such interventions are not possible with standard CBMs.*

**PGCMs are flexible and can work with a variety of segmentation methods**, showing that any type of segmentation can be used, depending on what is available. For ColorM-NIST+, we train a segmenter *jointly* with the network using the provided ground-truth masks. In CelebA, we *pretrain* a segmenter using the dataset's masks before using it to train the model. For CLEVR-Hans3, since ground-truth masks are not provided in the original dataset, we use the *Segment Anything Model (SAM)* (Kirillov et al., 2023) to extract object segmentations. For CUB, we consider the entire input

image as one image part, avoiding segmentation altogether, which effectively turns the model into a prototype network instead of a part-prototype network.

**PGCMs are more or equally responsive to concept interventions compared to standard CBMs (Figure 4).** In PGCMs, concepts are not modeled conditionally independent, which allows a single concept intervention to influence multiple concept predictions (*dashed line*, PGCM*). Figure 4 shows that this improves intervenability compared to CBMs and variants that model concepts independently for ColorMNIST+ (weak concept correlations) and especially for CUB (strong concept correlations). For CelebA, we notice that standard interventions perform better (PGCM), as the inter-concept dependencies are insufficient in this dataset. We include task accuracy intervention curves in Appendix D.

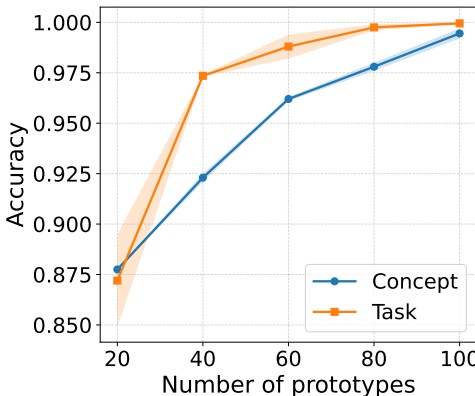

*Figure 5.* Concept and task accuracy on CLEVR-Hans3 for different numbers of learned prototypes.

**PGCMs' accuracy depends on model capacity, controlled by the number of learned prototypes like other prototype-based models (Figure 5).** Increasing the number of prototypes improves concept accuracy, which in turn leads to gains in task performance. However, as the number of prototypes grows, interpretability decreases due to higher cognitive load. When the number of prototypes is limited, the model may struggle to represent the full variability of

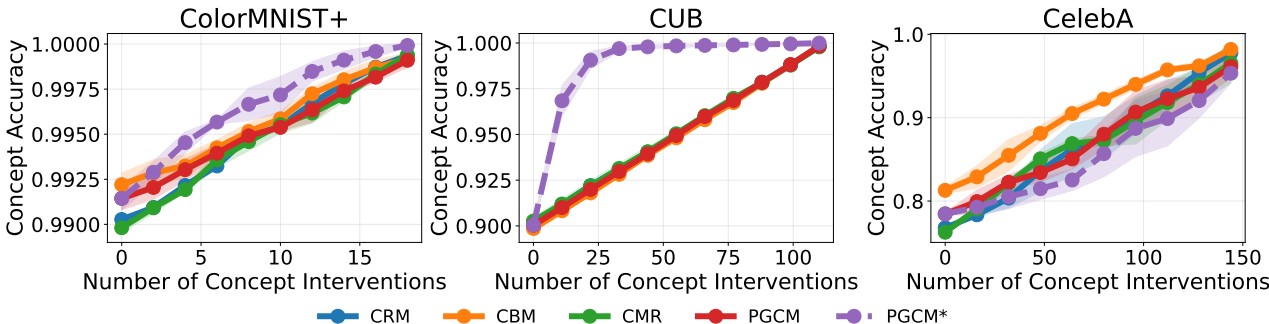

*Figure 4.* Concept accuracy after intervening on increasingly more concepts on ColorMNIST+, CUB, and CelebA. PGCM uses standard concept interventions, while PGCM* uses the interventions described in Section 4.3.

the data, resulting in reduced concept accuracy and, consequently, lower task accuracy. This highlights a fundamental trade-off in PGCMs between interpretability and capacity, similar to other prototype-based approaches.

## 7. Conclusion

We introduced *Prototype-Grounded Concept Models* (PGCMs), a new class of interpretable models that resolve a core and previously unaddressed limitation of Concept Bottleneck Models (CBMs): the inability to verify whether learned concepts align with their intended human semantics. By grounding each concept in a set of learned visual prototypes, PGCMs make concept meaning explicit, inspectable, and actionable. This replaces the strong and often unjustified assumption of concept alignment in CBMs with a concrete, verifiable mechanism. PGCMs retain the key advantages of CBMs (explicit concepts, transparent concept-to-task mappings, and concept interventions) while substantially expanding interpretability and intervenability. Through the concept alignment table, users can directly inspect the visual evidence underlying each concept, identify misalignments, and intervene by editing, removing, or adding prototypes.

By improving interpretability and enabling verification of concept alignment, the proposed PGCMs could increase trust, transparency, and human oversight in high-stakes AI applications such as healthcare or scientific analysis. A potential negative societal impact is that the same transparency and controllability could be used to intentionally steer model behavior in ways that enforce biases in the model.

**Limitations.** The prototype-based nature of PGCMs and the Concept Alignment Table introduces a trade-off between model capacity and interpretability. While increasing the number of prototypes can boost accuracy, it also increases the cognitive load for human verification. Moreover, the model's reliance on image parts means that performance is partially tied to the quality of the chosen segmentation method. However, our experiments show that the framework is highly flexible, performing well across various segmenters. Finally, a possible risk that PGCMs share with other CBMs that model concepts jointly is that a single intervention may wrongly affect other concept predictions.

## Impact statement

This work aims to advance machine learning by improving the interpretability and alignment of concept-based models. By making learned concepts explicit and inspectable, the proposed approach may support more transparent and trustworthy use of machine learning systems.

## Acknowledgements

This research has received funding from the KU Leuven Research Fund (GA No. STG/22/021), from the Research Foundation-Flanders FWO (GA No. G047124N, G033625N) and from the Flemish Government under the "Onderzoeksprogramma Artificiële Intelligentie (AI) Vlaanderen" programme. DD is a fellow of the Research Foundation-Flanders (FWO-Vlaanderen, 1185125N). PB is a fellow of the Swiss National Science Foundation (SNF grant 224226). We thank Lucas Van Praet for his insightful comments on the methodology.

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

# A. Additional model details

## A.1. Implementation

We implemented the Prototype-Grounded Concept Model (PGCM) using the PyTorch Lightning framework. The model architecture consists of the following primary components: a segmentation network, an image encoder, a set of learnable prototypes, a prototype decoder, a concept predictor and a task predictor.

**Architecture and Forward Pass.** The input image is first processed by a segmentation network. We employ a U-Net-based architecture with a ResNet backbone to predict $n$ segmentation masks, where $n$ corresponds to the number of image parts. Each extracted image part is mapped to a latent embedding using a shared Convolutional Neural Network (CNN) encoder. To determine the prototype selection distribution $q(S_i|X_i)$, we compute the similarity between the image part embedding and the learnable embeddings of all $m$ prototypes. For CUB, this is a cosine similarity between the part embedding and prototype embeddings, followed by a softmax operation to obtain a categorical distribution over prototypes with a temperature of 25. For all other datasets, we use dot product as similarity.

We do not directly use the learnable prototype embeddings (see Appendix A.2). Instead, during the forward pass, we first map the prototype embeddings to the image space using the image decoder. These generated prototype images are then re-encoded into latent embeddings using the shared image encoder (the same encoder used for input image parts). The concept predictor, implemented as an MLP, then takes these re-encoded embeddings as input to predict the concept probabilities for each prototype. For a specific input image part, the final concept prediction is computed as the expectation of these prototype concepts under the selection distribution $q(S_i|X_i)$: effectively a weighted average based on the similarity between the input part and the prototypes. Finally, the task predictor $p(Y|C)$ takes the aggregated concept probabilities (thresholded, to have hard concepts) across all image parts to predict the final task label.

To address class imbalance, we calculate positive class weights based on the frequency of each concept in the training dataset; these weights are applied specifically to the positive targets during the binary cross-entropy calculation for the concept loss. We also apply this weighting strategy to all competitor models to ensure a fair comparison.

Additionally, we include weights on the other terms of the ELBO (specifically, the reconstruction term and the entropy term) to balance the different objectives.

## A.2. Interpretability optimizations

**Mapping learned prototypes to concrete training instances.** After half of the training epochs, we replace the learned prototypes with concrete training instances. This step improves interpretability by ensuring that each prototype corresponds exactly to a ground truth image region from the training data, and should thus be semantically meaningful. Concretely, for each learned prototype embedding $e_j$, we identify the most similar image part $x_i$ in the training set by finding the highest dot product between their latent representations, $e_j$ and $f_{enc}(x_i)$. We then substitute the learned prototype with the selected image part $x_j$.

Following this replacement, we discard the learned prototype embeddings $\{e_j\}_j$, only using the predicted embedding $f_{enc}(x_j)$, as they are no longer necessary. The prototype images are frozen for the remainder of training, while all other components of the model continue to be optimized.

## A.3. Computational complexity

Computing the ELBO during training is $\mathcal{O}(n \cdot m) + \mathcal{O}(|y| \cdot k) + \mathcal{O}(n \cdot m \cdot k) + \mathcal{O}(n \cdot m \cdot D)$ with $n$ the number of image parts, $m$ the number of prototypes, $|y|$ the number of tasks, $k$ the number of concepts and $D$ the number of pixels in an input image. At test time, the complexity is $\mathcal{O}(n \cdot m \cdot (k + |y|))$.

# B. Experimental and implementation details

**Datasets.** For our experiments, we utilized the CelebA-HQ dataset, leveraging its provided segmentation masks to define four specific objects: skin, hair, lips, and nose. For each object, we selected a subset of relevant concepts that can be visually inferred from that specific region. The final downstream task consists of predicting three high-level attributes: "Young", "Attractive", and "Male". For ColorMNIST+, we started from Manhaeve et al. (2018)'s MNIST+ dataset, but color each digit either red, green or blue with equal probability. Each instance has two ground truth masks: one per MNIST digit.

In the noisy version of ColorMNIST+, we change the label of each instance of a 3 and a 4 to a 1 and an 8, respectively, with a probability of 30% (in the training and validation set). For the experiments in CLEVR-Hans, we followed the instructions and classes provided in the original dataset. To extract object segmentations, we used the pretrained Segment Anything Model (Kirillov et al., 2023), using bounding boxes as prompts. For CUB, we first finetune a ResNet18 (taken from Vandenhirtz et al. (2024)), where we add a linear layer with a softmax for predicting the classes and a linear layer with a sigmoid for predicting the concepts. We use a learning rate of 0.01 (SGD optimizer, momentum of 0.9), batch size 128 and train for 200 epochs. This is similar to the configuration many other CBM works use (Espinosa Zarlenga et al., 2022). We then drop the classification layers and run the CUB images through the ResNet18 to obtain image embeddings, which we use to train the PGCM and all competitors on (instead of on the images). For PGCM, this means our reconstruction loss reconstructs these input embeddings rather than the original images.

**Reproducibility.** We used seeds 1, 2 and 3 for all our experiments. We run all our experiments on multiple machines with a Nvidia L40S 48GB GPU card with AMD EPYC 9334 CPU and 256GB RAM. Small deviations in results are due to the different machine settings. We estimate the total time to run our experiments across all datasets and models is 150h. Experiments require significantly different training times depending on the complexity of the images and any masking and encoding methods used. Specifically, MNIST+ requires 41 minutes per run, CelebA 13 hours, CLEVR-Hans 5 hours, and CUB 40 minutes. We also estimate that including all the experiments required for the entire research project would double the total GPU time.

**Intervention policy.** We use a random policy by generating a random concept ordering and following it for all instances.

**General training information.** We used the AdamW optimizer with a learning rate schedule that features a linear warmup for the first 10 epochs, followed by a cosine annealing decay that gradually reduces the learning rate to a minimum of $10^{-5}$. To make the models more responsive to concept interventions, we incorporate Espinosa Zarlenga et al. (2022) *randint*, also performing concept interventions at random during training. Each concept prediction has a 20% probability to be intervened on.

**General architectural details of competitors.** To ensure fair comparison, we use the same number of layers and number of neurons within the layers for our model and all competitors. We use a Convolutional Neural Network for mapping the input image (or image part) to an embedding, from which we can continue with a multilayer perceptron (MLP). For CBM, the MLP predicts the concepts. For CRM, the MLP predicts the concepts and an additional residual embedding that is passed to the task predictor. For CMR, it predicts the concepts and the logits of CMR's categorical rule selector. For CBM, CRM and PGCM, the task predictor is another MLP. For CMR, it is CMR's standard rule-based task predictor.

**Number of learned prototypes.** For ColorMNIST+, we learn $m = 30$ prototypes. For CelebA, we learn $m = 120$ prototypes. The CLEVER-Hans experiment achieved the best result with $m = 100$ prototypes. For CUB, we learn $m = 350$ prototypes.

## C. Loss derivation

In this section, we derive that our PGM (Figure 3) allows the following ELBO for the likelihood:

$$\log p(\{x_i\}_i, \{c_i\}_i, y) \geq \underbrace{-\sum_i \mathrm{KL}\big(q(s_i|x_i) \,\|\, p(s_i)\big)}_{\text{regularization term}} + \underbrace{\log p(y \mid \{c_i\}_i)}_{\text{task loss}}$$
$$+ \sum_i \left( \mathbb{E}_{q(s_i|x_i)}\Big[ \underbrace{\log p(c_i \mid s_i)}_{\text{concept loss}} + \underbrace{\log p(x_i \mid s_i)}_{\text{reconstruction loss}} \Big] \right) \tag{5}$$

We start from the likelihood and marginalize the unobserved variables $S_i$:

$$\log p(\{x_i\}_i, \{c_i\}_i, y) = \log \sum_{\{s_i\}_i} p(\{s_i\}_i) \cdot p(\{x_i\}_i, \{c_i\}_i, y | \{s_i\}_i) \tag{6}$$

with $m$ the number of prototypes and $n$ the number of image parts. We introduce a variational posterior $q(\{s_i\}_i | \{x_i\}_i) =$

$\prod_i q(s_i|x_i)$ and use Jensen's Inequality.

$$\log p(\{x_i\}_i, \{c_i\}_i, y) = \log \sum_{\{s_i\}_i} p(\{s_i\}_i) \cdot p(\{x_i\}_i, \{c_i\}_i, y|\{s_i\}_i) \cdot \left( \frac{q(\{s_i\}_i|\{x_i\}_i)}{q(\{s_i\}_i|\{x_i\}_i)} \right) \tag{7}$$

$$\geq \sum_{\{s_i\}_i} q(\{s_i\}_i|\{x_i\}_i) \cdot \log \left( p(\{x_i\}_i, \{c_i\}_i, y|\{s_i\}_i) \cdot \left( \frac{p(\{s_i\}_i)}{q(\{s_i\}_i|\{x_i\}_i)} \right) \right) \tag{8}$$

We then split the logarithm:

$$\log p(\{x_i\}_i, \{c_i\}_i, y) \geq \sum_{\{s_i\}_i} q(\{s_i\}_i|\{x_i\}_i) \cdot \log \left( \frac{p(\{s_i\}_i)}{q(\{s_i\}_i|\{x_i\}_i)} \right) \tag{9}$$

$$+ \sum_{\{s_i\}_i} q(\{s_i\}_i|\{x_i\}_i) \cdot \log p(\{x_i\}_i, \{c_i\}_i, y|\{s_i\}_i) \tag{10}$$

We exploit the conditional independencies encoded by the PGM to simplify the first term. In particular, the variational posterior and prior factorize as $q(\{s_i\}_i \mid \{x_i\}_i) = \prod_i q(s_i \mid x_i)$ and $p(\{s_i\}_i) = \prod_i p(s_i)$. Therefore,

$$\sum_{\{s_i\}_i} q(\{s_i\}_i \mid \{x_i\}_i) \log \frac{p(\{s_i\}_i)}{q(\{s_i\}_i \mid \{x_i\}_i)} = \sum_{\{s_i\}_i} \left( \prod_j q(s_j \mid x_j) \right) \log \frac{\prod_i p(s_i)}{\prod_i q(s_i \mid x_i)} \tag{11}$$

$$= \sum_{\{s_i\}_i} \left( \prod_j q(s_j \mid x_j) \right) \sum_i \log \frac{p(s_i)}{q(s_i \mid x_i)} \tag{12}$$

$$= \sum_i \sum_{\{s_i\}_i} \left( \prod_j q(s_j \mid x_j) \right) \log \frac{p(s_i)}{q(s_i \mid x_i)} \tag{13}$$

$$= \sum_i \sum_{s_i} q(s_i \mid x_i) \log \frac{p(s_i)}{q(s_i \mid x_i)} \tag{14}$$

$$= - \sum_i \mathrm{KL}\big(q(s_i \mid x_i) \,\|\, p(s_i)\big). \tag{15}$$

The fourth equality follows because each logarithmic term depends only on $s_i$, and all remaining latent variables marginalize to one under the factorized variational posterior.

We now simplify the second term. Using the conditional independencies encoded by the PGM, we have:

$$p(\{x_i\}_i, \{c_i\}_i, y \mid \{s_i\}_i) = p(y \mid \{c_i\}_i) \prod_i p(c_i \mid s_i) \prod_i p(x_i \mid s_i). \tag{16}$$

Taking the logarithm and substituting into the expectation yields:

$$\sum_{\{s_i\}_i} q(\{s_i\}_i \mid \{x_i\}_i) \cdot \log \ p(\{x_i\}_i, \{c_i\}_i, y \mid \{s_i\}_i) \tag{17}$$

$$= \sum_{\{s_i\}_i} q(\{s_i\}_i \mid \{x_i\}_i) \Big[ \log p(y \mid \{c_i\}_i) + \sum_i \log p(c_i \mid s_i) + \sum_i \log p(x_i \mid s_i) \Big] \tag{18}$$

$$= \log p(y \mid \{c_i\}_i) + \sum_i \sum_{\{s_i\}_i} q(\{s_i\}_i \mid \{x_i\}_i) \big[ \log p(c_i \mid s_i) + \log p(x_i \mid s_i) \big] \tag{19}$$

$$= \log p(y \mid \{c_i\}_i) + \sum_i \sum_{s_i} q(s_i \mid x_i) \big[ \log p(c_i \mid s_i) + \log p(x_i \mid s_i) \big] \tag{20}$$

$$= \log p(y \mid \{c_i\}_i) + \sum_i \mathbb{E}_{q(s_i|x_i)} \Big[ \log p(c_i \mid s_i) + \log p(x_i \mid s_i) \Big]. \tag{21}$$

Combining both terms, we obtain the following evidence lower bound (ELBO):

$$
\begin{aligned}
\log p(\{x_i\}_i, \{c_i\}_i, y) \geq &- \sum_i \mathrm{KL}\big(q(s_i \mid x_i) \,\|\, p(s_i)\big) + \log p(y \mid \{c_i\}_i) \\
&+ \sum_i \mathbb{E}_{q(s_i \mid x_i)}\Big[ \log p(c_i \mid s_i) + \log p(x_i \mid s_i) \Big].
\end{aligned}
\tag{22}
$$

## D. Additional Results and Visualisations

We did not evaluate the competitors on CLEVR-Hans, as object-centric datasets require set-matching procedures to compute loss and accuracy. In our approach, this additional step is unnecessary due to the explicit segmentation stage. Nevertheless, we expect that these competitors would achieve near-perfect accuracy on this dataset when combined with set-matching, similar to our method.

Figure 6 shows how task accuracy evolves as a function of the number of concept interventions across datasets (CUB, ColorMNIST+, CelebA). As expected, competitors with a sidechannel (CMR, CRM) show limited responsiveness to interventions, as their task predictions do not rely solely on the concept bottleneck. On ColorMNIST+ (weak concept correlations) and especially CUB (strong concept correlations), the extended intervention mechanism (PGCM*) substantially outperforms all baselines: because interventions operate through prototypes, a single correction can simultaneously fix multiple correlated concepts (see Figure 4), leading to larger improvements in task accuracy. This compounding effect becomes more pronounced as additional interventions are applied. On CelebA, where concept interventions have only a weak effect on downstream predictions overall, improvements remain modest for all methods. In this regime, PGCM behaves similarly to other CBMs (task accuracy increasing slightly), while PGCM* can even slightly underperform due to the weak correlations.

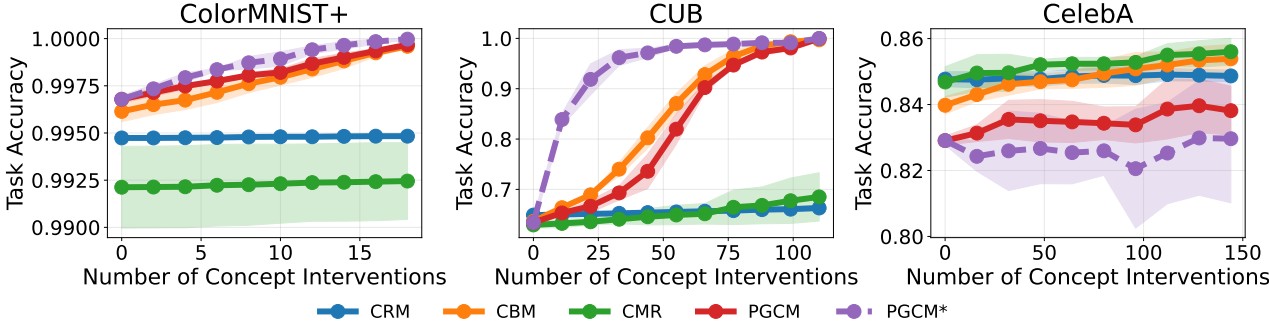

*Figure 6.* Task accuracy after intervening on increasingly more concepts on ColorMNIST+, CUB and CelebA. PGCM uses standard concept interventions, while PGCM* uses the interventions described in Section 4.3.

*Figure 7.* Visualization of model outputs on CelebA, comparing our PGCM with competing CBM-based approaches. For each input image, we show the predicted concepts and the corresponding visual evidence used by the models. While CBM-based methods only provide concept predictions, PGCM additionally grounds these predictions in selected prototypes, making the underlying visual reasoning explicit.

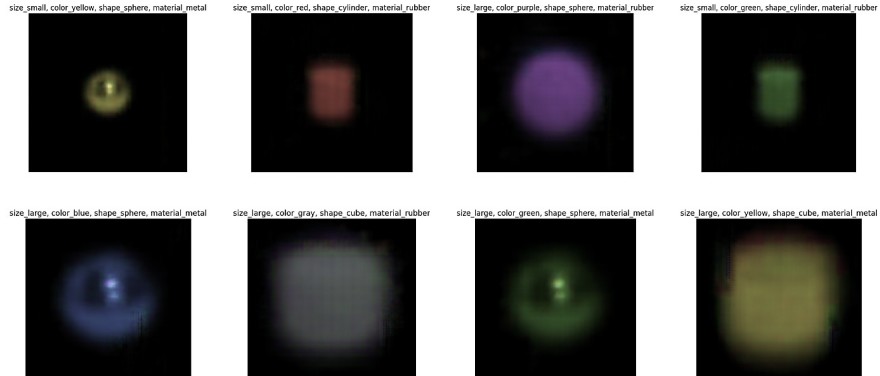

*Figure 8.* Generated prototypes on CLEVR-Hans3 together with their predicted concept representations during training, before the prototype swapping step. At this stage, prototypes are still learned representations and may not correspond to realistic image parts. The figure shows how prototypes already capture meaningful visual patterns and associated concepts, even before being replaced by nearest training instances.

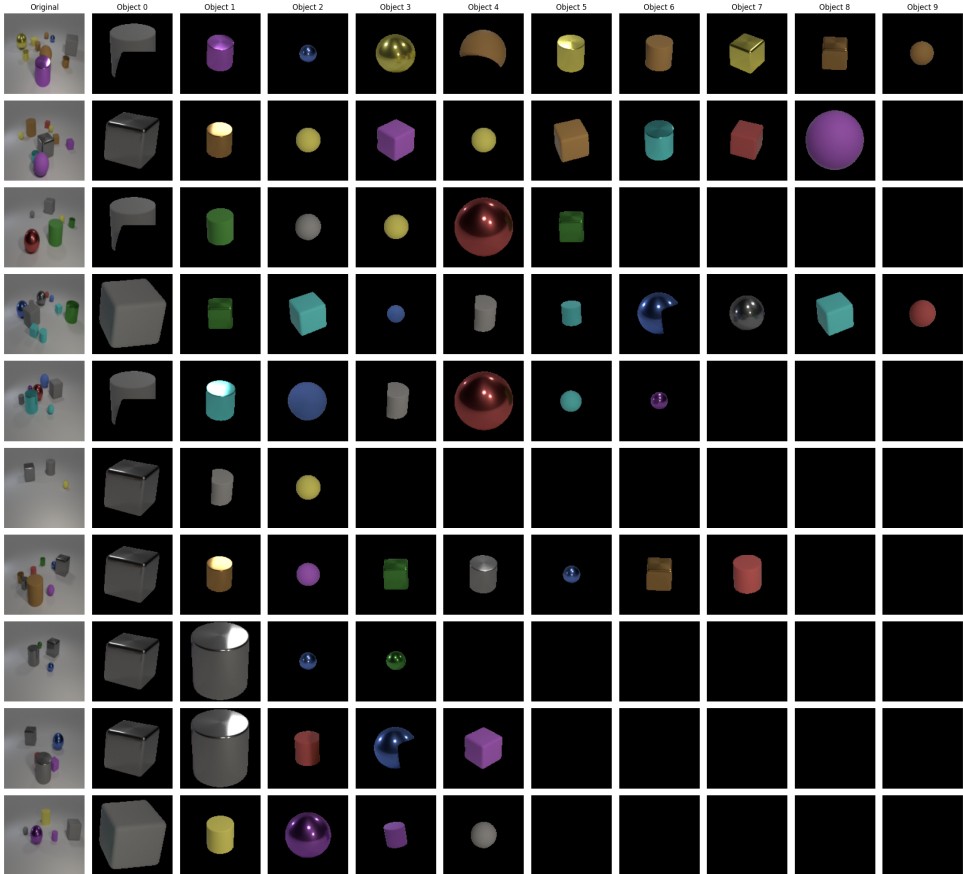

*Figure 9.* Examples from the CLEVR-Hans3 validation set together with the prototypes selected by our model for each extracted image part. The displayed prototypes correspond to the final model after the prototype swapping step, and therefore represent real image parts from the training data. The figure illustrates how the model assigns prototypes to different objects in the scene; in cases where no additional object is present, the model selects an empty prototype (black image).

## E. Licenses

All the data we used to build our datasets are freely available on the web with licenses:

- MNIST+: CC BY-SA 3.0 DEED,

- CelebA: available for non-commercial research purposes only[2]

- CUB: MIT license

- CLEVR-Hans 3: MIT license

---

[2]https://mmlab.ie.cuhk.edu.hk/projects/CelebA.html

