# OpenReview forum: "Prototype-Grounded Concept Models for Verifiable Concept Alignment"
_ICML.cc/2026/Conference — ICML 2026 regular_

### Official Review · Reviewer_rWm8 · 2026-03-12

**Soundness:** 1
**Presentation:** 2
**Significance:** 3
**Originality:** 2
**Overall Recommendation:** 4
**Confidence:** 3

**Summary:**

The paper proposes Prototype-Grounded Concept Models (PGCMs), a combination of prototype networks and a CBM model. The core motivation is that standard CBMs provide no way to verify whether learned concepts align with their intended human semantics. PGCMs resolve this by grounding each concept in learned visual prototypes.

# Method

Define the prototype embeddings as $e_j$, randomly initialized and learned during training.

## During Training

- A segmentation network is used to extract image parts $x_i$ from image $X$. This can be either pretrained or learned, if learned, ground truth masks are required.

- Each image segment $x_i$ is passed to the encoder network $f_{enc}(x_i)$ to get an embedding $z_i$.

- The prototype selector computes prototype assignment probabilities using the dot product between every prototype embedding $e_j$ and the image part embedding $z_i$, combined with a softmax to obtain a categorical distribution $q(S_i | X_i)$.

- There is a network $f_{concept}$ that takes $e_j$ and gives concept probabilities for each prototype.

- $f_{concept}(e_j)$ is multiplied with the prototype selector distribution $q(S_i | X_i)$ (weighted average) to get the probability of a given concept for an image part. A cross-entropy concept loss is added on top of this.

- The predicted concept probabilities are then passed to a linear layer (following the CBM design) to get the task prediction. A task loss is applied here.

- There is a network $f_{image}$ that takes $e_j$ and reconstructs the image, to which a reconstruction loss is applied.

- There is a KL regularization term between the prototype selector $q(S_i | X_i)$ and a fixed uniform categorical distribution over all $m$ prototypes $p(S_i)$, preventing the model from collapsing onto a few prototypes.

- To ensure prototype embeddings cannot encode information beyond what is visually present in their image representations, PGCM compute $e_j = f_{enc}(f_{image}(e_j))$ and use $e'_j$ instead of $e_j$ everywhere. This ensures prototypes map to real images.

- This gives us a concept alignment table where each $e_j$ has both an image representation and a concept distribution mapped to it.

## During Inference

- Segment $X$ to get image parts $x_i$
- Encode each $x_i$ via $f_{enc}$ to get $z_i$
- Take $\arg\max$ of dot product between $z_i$ and all $e_j$ to select a prototype
- Read off the concepts associated with the selected prototype from the concept alignment table
- Pass concepts through the linear layer to make the task prediction $\hat{Y}$


# Interpretability
Since predictions are made from masked image parts (extracted by the segmenter), the model gives you a formal guarantee that the rest of the image was not used for that prediction. PGCM allows for the following interventions:

- Prototype-level interventions (alignment/model editing): edit, remove, or add prototypes in the concept alignment table.
- Concept-level interventions: same as standard CBMs, correcting a mispredicted concept at test time.
- Prototype selection intervention:  at test time, you can override which prototype gets selected for a given image part.
# Experiments


-  **Datasets:** The paper use two datasets CelebA and ColorMNIST+
-  **Interpretability and Alignment**: The paper first look at the concept alignment tables to check what concepts are active for each prototype and whether the semantics make sense visually.
- **Model Editing**: To evaluate model editing they give ColorMNIST+ noisy concept labels during training so the model learns wrong concept associations and some prototypes end up with wrong labels. They then show that by dropping the wrong prototypes or editing their concept labels accuracy on the non-noisy test set significantly increases.
- **Accuracy**: On CelebA PGCM get slightly lower concept and task accuracy compared to baselines.
- **Interventions**: On CelebA standard PGCM interventions perform slightly worse than baselines.

**Compliance With Llm Reviewing Policy:**

Affirmed.

**Final Justification:**

Based on the rebuttals, the author did add an additional dataset which addresses my issue on scale so I increased the score accordingly however, I am not fully convinced that other baselines are in applicable in this case.

**Key Questions For Authors:**

**Q1:** What is the effect of using a pretrained segmentation network versus one learned from scratch? How sensitive is the model to the quality of the segmentation, and does a better segmenter directly translate to better concept alignment?

**Q2:** How is the number of prototypes $m$ chosen, and how sensitive is the model to this hyperparameter? Specifically, does increasing $m$ recover the accuracy gap observed on CelebA, and is there a tradeoff between the number of prototypes and interpretability?

**Q3:** How does PGCM scale to datasets with a larger and more diverse concept set such as CUB? CelebA already shows a noticeable accuracy drop compared to baselines, which raises concerns about applicability to more complex real-world settings.

**Q4:** Prior work such as [1] and [2] also combines visual prototypes with CBMs. How does PGCM  improve upon these works, and why are they not included as baselines? Without this comparison it is difficult to assess what the probabilistic framework contributes beyond simpler existing approaches. A clear differentiation and empirical comparison would significantly change our evaluation.

## References
- [1] Aligning Visual and Semantic Interpretability through Visually Grounded Concept Bottleneck Models, 2024
- [2] DCBM: Data-Efficient Visual Concept Bottleneck Models, 2025

**Strengths And Weaknesses:**

## Significance: Good

**Strengths:**
- The core idea of grounding concepts in visual prototypes is genuinely useful for interpretability. It allows direct visual inspection of what the model actually learned for each concept, making spurious correlations easy to detect.
- PGCMs introduce three levels of intervention, prototype-level model editing, standard concept-level interventions, and prototype selection interventions, which is a real improvement over standard CBMs that only support concept-level interventions.

**Weaknesses:**
- The model involves a large number of networks trained jointly: $f_{enc}$, $f_{image}$, $f_{concept}$, the segmentation network, the task predictor, and the prototype embeddings $e_j$, which raises concerns about training stability and scalability to larger datasets.

## Originality: Fair

**Strengths:**
- The probabilistic framework (ELBO combining reconstruction, concept, task, and KL losses) and the dual representation per prototype (image + concept) is a clean and principled way to combine prototype networks and CBMs.

**Weaknesses:**
- The combination of visual concepts and CBMs has been done before in prior work [1, 2]. The paper does not cite these works or explain how PGCMs differ from them, and does not use them as baselines.

## Presentation: Fair

**Weaknesses:**
- The flow of Section 3 is hard to follow. The natural way to understand the model is to follow the forward pass: $X \rightarrow x_i \rightarrow z_i \rightarrow S_i \rightarrow C \rightarrow Y$. However, the section is structured around the probabilistic graphical model, introducing the generative model (image decoder, concept decoder) before the discriminative path (encoder, prototype selector) that is actually used for inference. This makes it unnecessarily hard to build an intuition for the model on a first read.

## Soundness: Poor

**Weaknesses:**
- The empirical results are weak. ColorMNIST+ is essentially a toy task, so the only real dataset is CelebA, where PGCMs underperform all baselines in both concept accuracy and task accuracy. The paper attributes this to the capacity constraint of a finite prototype set, but does not provide sufficient analysis or ablations to support this claim or suggest how it could be addressed.
- The baselines are not sufficient. Given that combining visual prototypes and CBMs has been done in prior work [1, 2], these should be included as baselines.
- The core claim of the paper is that PGCMs improve interpretability and concept alignment verification. However this claim is only backed by a few qualitative examples in Tables 1 and 2. There is no user study, no quantitative interpretability metric, and no systematic evaluation of whether humans can actually verify concept alignment better using PGCMs than standard CBMs.


## References
- [1] Aligning Visual and Semantic Interpretability through Visually Grounded Concept Bottleneck Models, 2024
- [2] DCBM: Data-Efficient Visual Concept Bottleneck Models, 2025

---

> ### Author Rebuttal · Authors · 2026-03-30
>
> We are glad the reviewer considers our idea genuinely useful for interpretability, our introduced interventions a real improvement over standard CBMs, and our probabilistic framework a clean and principled way to combine prototypes and CBMs.
>
> &nbsp;
>
> > The model involves a large number of networks trained jointly which raises concerns about training stability and scalability to larger datasets.
>
> **We did not observe any issues with training stability** in our experiments.
>
> Moreover, **several components can be pretrained independently or trained sequentially** (e.g. segmenter, task predictor).
>
> &nbsp;
>
> > The combination of visual concepts and CBMs has been done before [1, 2]. The paper does not cite them. How does PGCM improve upon them, and why are they not included as baselines?
>
> We will discuss them in the related work.
>
> [1] and [2] combine visual prototypes with CBMs, but **they do concept discovery rather than operating over a predefined supervised concept set**. As a result, the human cannot enforce or verify alignment with a given set of concepts. **Our goal is not to discover concepts, but to make predefined concepts' alignment verifiable.**
>
> For this reason, **these methods are not directly comparable baselines, as they address a different problem setting.**
>
> &nbsp;
>
> > What is the effect of using a pretrained segmentation network vs a learned one? How sensitive is the model to the segmentation quality, and does a better segmenter translate to better concept alignment?
>
> **In our work, segmentation must be seen similarly to concept supervision: it is fundamental for alignment.** Better segmentation quality leads to more coherent parts and thus cleaner prototypes, which can improve concept alignment. **The key advantage of our approach is that any alignment issues are directly observable**: if alignment is poor, this will be reflected in the learned prototypes. This cannot be detected in existing CBMs.
>
> **We conducted an additional experiment on CLEVR-Hans3 using a pretrained segmenter.** The model achieves strong performance (99.2% concept acc.).
>
> &nbsp;
>
> > The empirical results are weak. ColorMNIST+ is essentially a toy task, so the only real dataset is CelebA, where PGCMs underperform in concept and task accuracy. The paper attributes this to the capacity constraint of a finite prototype set, but does not provide ablations to support this.
>
> > How is the number of prototypes m chosen, and how sensitive is the model to this? Specifically, does increasing m recover the accuracy gap observed on CelebA, and is there a tradeoff between the number of prototypes and interpretability?
>
> As with all prototype networks, **'m' is a user-defined hyperparameter that reflects an accuracy-interpretability trade-off**: increasing 'm' improves model capacity and performance, but also increases the cognitive load.
>
> **We conducted an additional experiment where we increase 'm'.** With 120 prototypes, PGCM achieves a performance similar to CBMs on CelebA (78.6% concept acc.).
>
> **To further show the sensitivity, we conducted an extra experiment on CLEVR-Hans3** with a pretrained segmenter. We report concept accuracy when learning different numbers of prototypes 'm':
> - m=100: 99.2%
> - m=80: 98.0%
> - m=60: 96.1%
> - m=40: 92.5%
> - m=20: 87.7%
>
> &nbsp;
>
> > How does PGCM scale to datasets with a larger and more diverse concept set such as CUB? CelebA already shows an accuracy drop compared to baselines
>
> **Scalability to larger concept sets is primarily a matter of model capacity.** Datasets with more concepts would in principle demand a larger number of prototypes. However, our goal is to maintain interpretability, which limits the number of prototypes that can be used (depending on the human). Increasing that number improves accuracy, while restricting it leads to better interpretability. This trade-off is clarified by the additional experiment described in the answer above.
>
> &nbsp;
>
> > The core claim of the paper is that PGCMs improve interpretability and concept alignment verification. However this claim is only backed by a few qualitative examples in Tables 1 and 2. There is no user study, no quantitative interpretability metric, and no systematic evaluation of whether humans can actually verify concept alignment better using PGCMs than standard CBMs
>
> **In existing CBMs, concept prediction is fully black-box, making alignment _impossible_ to verify.**
>
> **Our claim** that concept alignment can be inspected by the human **directly follows from grounding them in prototypes.** Our main contribution is not to show aligned models; it is to enable verification of concept alignment. This is architectural rather than experimental. In this sense, we believe more qualitative examples or user studies would not contribute to this.
>
> &nbsp;
>
> > The flow of Section 3 is hard to follow. It introduces the generative model before the discriminative path that is actually used for inference.
>
> We can definitely present the discriminative path first.

---

> > ### Author Rebuttal · Reviewer_rWm8 · 2026-04-04
> >
> > I thank the authors for their detailed rebuttal. However, my two main concerns remain unresolved:
> > (a) The missing baseline comparison is still a significant gap. I disagree with the authors' claim that prior works combining visual prototypes with CBMs [1, 2] are not comparable,  these methods also ground concepts visually and can be inspected by humans, making them natural baselines. This concern has also been raised by reviewer uqfG, who similarly questioned whether the distinction between latent-space and image-space prototypes is as fundamental as the authors suggest.
> > (b) The additional CLEVR-Hans3 experiment does not adequately address my concern about real-world applicability. As also noted by reviewer pmPd, CLEVR-Hans3 is a synthetic dataset, and results on it do not demonstrate that the method generalizes to complex, real-world scenarios. Furthermore, the experiment does not address whether the approach scales to datasets with larger and more diverse concept sets such as CUB or AWA2, where the accuracy gap observed on CelebA may widen further.
> > For these reasons, I am maintaining my score.

---

> > > ### Author Response · Authors · 2026-04-07
> > >
> > > > (a) The missing baseline comparison is still a significant gap. I disagree with the authors' claim that prior works combining visual prototypes with CBMs [1, 2] are not comparable
> > >
> > > [1, 2] do concept discovery. They do not learn the same concept set as PGCMs or CBNMs. Thus, **they cannot be compared regarding concept accuracy** (which is the main comparison point of our paper, given that our method is about concept prediction).
> > >
> > > &nbsp;
> > >
> > > > Reviewer uqfG, who similarly questioned whether the distinction between latent-space and image-space prototypes is as fundamental as the authors suggest
> > >
> > > **Without a faithful image representation of a prototype, it is impossible to verify concept alignment.** Existing part-prototype networks do not learn image-representations of prototypes. The core problem is that they interpret prototypes in a post-hoc manner, which is insufficient for verifying alignment. This is fundamental: if it were not, we would have used existing prototype networks to predict the concepts. **Because existing part-prototype networks are insufficient, we had to develop a new one.**
> > >
> > > We refer to our final response to Reviewer uqfG to show that existing part-prototype networks do not learn image representations.
> > >
> > > Finally:
> > > - Comparing to them regarding learned prototypes is not useful; irrespective of which prototypes they learn, their prototype learning is not a competitor to our prototype learning because they do not learn image representations (see above).
> > > - Comparing to them regarding concept accuracy is not useful, given the above. Even if they are more accurate, they could not be used for our purpose. The important research question is: how much accuracy does the ability to verify concept alignment cost (with respect to Concept Bottleneck Models). Our experiments aim to answer this.
> > > - Comparing to them regarding task accuracy is not useful for the same reason: they cannot be used for our purpose.
> > >
> > > &nbsp;
> > >
> > > > (b) The additional CLEVR-Hans3 experiment does not adequately address my concern about real-world applicability. As also noted by reviewer pmPd, CLEVR-Hans3 is a synthetic dataset, and results on it do not demonstrate that the method generalizes to complex, real-world scenarios. Furthermore, the experiment does not address whether the approach scales to datasets with larger and more diverse concept sets such as CUB or AWA2, where the accuracy gap observed on CelebA may widen further.
> > >
> > > **We have performed an additional experiment on CUB** with a pretrained feature extractor (ResNet18). With 250 prototypes, we achieve 89.1% accuracy (vs 91.7% for the CBM), showing again PGCMs come with only have a small accuracy drop.

---

### Official Review · Reviewer_pmPd · 2026-03-13

**Soundness:** 3
**Presentation:** 3
**Significance:** 2
**Originality:** 3
**Overall Recommendation:** 3
**Confidence:** 3

**Summary:**

This paper introduces Prototype Grounded Concept Models (PGCMs), which extends a CBM to get more human-interpretable concepts. Since, a CBM inherently does not have a mechanism to verify if the model's internal representation of a concept can be interpreted by a human, PGCM tries to bridge this gap. The authors introduce a "concept alignment table" that collects concepts from the concept decoder with an image representation. Users can inspect what the model thinks a concept looks like and intervene if misalignment is detected. The model is formulated as a latent variable model with a variational inference objective (ELBO). Experiments are conducted on CelebA and ColorMNIST+ datasets and they show that PGCMs perform at similar levels of accuracy compared to standard CBMs and are similarly responsive to concept interventions as compared to CBMs and Table 4 demonstrates this conclusion.

**Compliance With Llm Reviewing Policy:**

Affirmed.

**Final Justification:**

The rebuttal has addressed some of my concerns on the choice of m, application of PGCM, and additional experimental results. The paper presents a good contribution to the CBM space. However, the result on generalizability beyond CelebA is not enough. I appreciate the results on CLEVR and a single experiment result on CUB but this might not give enough evidence as CelebA is the only real world dataset application. I also appreciate the authors explaining the application of PGCM and how it can be used to make concept prototypes better. I will maintain my score and I hope the authors can make the suggested changes and experiment on more real-world data to strengthen the contribution.

**Key Questions For Authors:**

- Can the authors show some preliminary results on other standard and widely used benchmarks such as CUB and AWA2 datasets? I believe positive results on these datasets can support the claims made in the paper.
- Are any human evaluations considered for the concept alignments to validate if the concepts do end up becoming more interpretable by humans? If it is not possible to get human evaluators, then even showing a side by side comparison can help visualize how PGCM are learning better concepts.
- Have you run any experiments where a pretrained segmenter was used instead of ground truth segmentation masks? What architecture could be used and how different the performance be as compared to the ground truth segmentations?

**Limitations:**

The paper does not discuss the limitations completely such as those on the datasets it experiments on and the scalability of the concept interventions

**Strengths And Weaknesses:**

**Strengths**

- The motivation seems to be clear that CBMs inherently don't have a way to verify if a learned concept actually means something in real life and to a human. The authors describe the problem setting in good detail using Figure 1 and this figure also highlights how PGCMs are different from other CBMs.
- I liked Section 3.2 of the paper, where the different parameterizations are described and makes it easier to understand the paper. The loss decompositions in Section 3.4 seems to be natural and each of the terms such as regularization loss, to give weightage to multiple protoypes instead of collapsing to a single prototype, task loss, which is the classification loss I think, concept loss, the supervised concept learning, and reconstruction loss, which helps the learned prototypes to be visually similar to the image.
- Another contribution that the paper makes apart from the PGCM model is the "concept alignment table", which helps map the concept to an image and also to a concept distribution, which I think makes the intervention of concept part easier as compared to letting a human judge a concept and then try to match it with a particular class of images. The authors also describe this intervention capability in Section 4.3, where a user can edit, remove or add learned prototypes, understand the inter-relation of concepts and intervene on prototype selection.
- The authors also describe the theoretical computational complexity of training and inference, and this is impressive as a lot of papers don't generally describe such computational complexities.

**Weaknesses**
- The authors only experiment with 2 datasets given in the paper, which are ColorMNIST+ and CelebA. Most CBM and concept related papers work with benchmark datasets like CUB and AWA2 datasets as these datasets have a rich set of annotations for different class of images. Results on these datasets could significantly strengthen the general applicability of the PGCMs and concept alignment table.
- On CelebA, PGCM achieves 75% concept accuracy which is much lower than the other methods, especially 81% on standard CBM, similarly PGCM achieves 81.6% task accuracy while 84% using CBMs. These are big gaps but the paper claims that PGCMs perform at a similar level compared to baselines mentioned in Table 4, and PGCMs are written to have more meangiful concepts than CBMs but if the accuracy drops so much then we cannot really say they are more meaningful. If the intervention improves this accuracy then the initial accuracy to actually be similar to the baseline and then the intervention improves the performance would be a better result.
- The paper also claims that the concepts are now more human interpretable and how users can inspect the concept alignment table to verify the alignment, but is a user study required here or is this a hypothetical benefit that could be achieved?
- The framework also uses some form of a segmentation mask, where these masks either come from the dataset or from a pretrained segmenter. How sensitive are these pretrained segmenters? The paper uses CelebA's own segmentations and for ColorMNIST+ a manual segmentation has been performed, so how does this change when pretrained segmenters are used instead?
- Eventhough I really like the concept alignment table, I don't see how this can be scaled up with a higher number of concepts if a user needs to manually intervene on the concepts. Is an automated way possible so that a user can efficiently intervene on the concepts and perform the interventions like add, edit and delete them?

---

> ### Author Rebuttal · Authors · 2026-03-30
>
> We are glad the reviewer believes we describe the problem setting in good detail with a clear motivation and clearly explained model.
>
> &nbsp;
>
> > (Question) Results on additional standard datasets such as CUB and AWA2 could significantly strengthen the general applicability of PGCMs.
>
> Due to the limited time in the rebuttal phase, we were unable to experiment on these datasets. However, **we did expand our evaluation to include the CLEVR-Hans3 dataset**, showing PGCM's ability to process multi-object scenes also **without ground-truth segmentation masks** (for results, see below). We specifically selected CLEVR-Hans3 because it required us to use a pretrained SAM model to extract the object segmentations.
>
> &nbsp;
>
> > On CelebA, PGCM achieves much lower concept accuracy than the other methods. But the paper claims that PGCMs perform comparable to baselines mentioned in Table 4, and PGCMs are written to have more meangiful concepts than CBMs but if the accuracy drops so much then we cannot really say they are more meaningful. If the intervention improves this accuracy then the initial accuracy to actually be similar to the baseline and then the intervention improves the performance would be a better result.
>
> As with all prototype networks, our accuracy is limited by the number of prototypes we learn (line 416+, col. 1). **We conducted an additional experiment where we increase the number of prototypes for CelebA (120)**, allowing PGCM to match more closely CBM performance on CelebA (78.6% concept accuracy). **We will change the claim** to make clear there is a small drop in accuracy.
>
> Our claim about “more meaningful concepts” does not rely on accuracy. PGCM allows the human to verify concept alignment, but other CBMs do not. **The human cannot know whether the learned concepts in existing CBMs are meaningful, but they can do this in PGCMs.**
>
> &nbsp;
>
> > The paper claims that the concepts are now more human interpretable and that users can inspect the concept alignment table to verify the alignment, but is a user study required here or is this a hypothetical benefit? If it is not possible to get human evaluators, then showing a side by side comparison can visualize how PGCM are learning better concepts.
>
> **Our claim is not that PGCMs learn perfectly aligned concepts more often than other CBMs.** Our contribution is that **alignment can be explicitly inspected, which is impossible for other CBMs.** This directly follows from representing concepts through prototypes. For this reason, **we believe a user study would provide no additional insight.**
>
> Moreover, we do not see how a side-by-side comparison is possible, as only PGCMs allow inspecting concept alignment; no visualisation is possible for CBMs. This is a clear advantage of PGCMs over CBMs.
>
> &nbsp;
>
> > You use segmentation masks, which either come from the dataset or from a pretrained segmenter. How sensitive are these pretrained segmenters? The paper uses datasets with masks, so how does this change when pretrained segmenters are used instead? What architecture could be used and how different would performance be?
>
> **We have added an experiment to show that PGCMs remain highly effective when utilizing pretrained segmenters**, achieving 99.2% concept accuracy and 99.9% task accuracy on CLEVR-Hans3 using SAM-generated masks.
>
> While our performance depends on the quality of the input masks (since consistently poor segmentation would corrupt the extracted visual parts) modern pretrained models resolve this bottleneck. To explicitly test this, we used the Segment Anything Model (SAM) on the CLEVR-Hans3 dataset (no ground-truth masks during training). The resulting accuracy shows that pretrained segmenters are robust enough to replace manual annotations without meaningful performance degradation.
>
> &nbsp;
>
> > Eventhough I really like the concept alignment table, I don't see how this can be scaled up with a higher number of concepts if a user needs to manually intervene on the concepts. Is an automated way possible so that a user can efficiently intervene on the concepts and perform interventions like add, edit and delete them?
>
> We agree that manual intervention becomes more challenging as the number of concepts grows. More broadly, **this limitation is common to all CBMs**: CBMs with many concepts are inherently difficult to interpret.
>
> For model interventions, automated approaches could be used, e.g. leveraging vision-language models to suggest or perform model edits. Still, human intervention remains the most reliable way to ensure correct alignment, and our method is the only one that makes this possible.
>
> &nbsp;
>
> > The paper does not discuss the limitations completely such as those on the datasets it experiments on and the scalability of the concept interventions
>
> Regarding scalability, concept interventions have the same computational complexity as in standard CBMs.
>
> We kindly ask the reviewer to clarify which limitations on the datasets they refer to.

---

> > ### Author Rebuttal · Reviewer_pmPd · 2026-04-02
> >
> > I thank the authors for their detailed answers. Some of my concerns have been partially addressed and I also appreciate the additional experiments conducted. My concern regarding scalability and the experiment on CLEVR is satisfactory, but CLEVR is a synthetic dataset and it does not completely resolve my concern.
> >
> > I appreciate the new experiment with 120 prototypes on CelebA. I had a followup question that how can a user set the number of prototypes m, if there is a systematic method apart from trial and error?
> >
> > Regarding the interpretability of the prototypes, the paper claims that PGCMs make the concepts more interpretable to humans. If a person tries to use the concept alignment table to intervene on the concepts then it is the user's ability and not the model design, even a small form of human evaluation or comparison between the prototypes and the CBM concept activations might help.
> >
> > I am maintaining my current score but I am willing to reconsider my score depending on the responses.

---

> > > ### Author Response · Authors · 2026-04-07
> > >
> > > > My concern regarding scalability and the experiment on CLEVR is satisfactory, but CLEVR is a synthetic dataset and it does not completely resolve my concern.
> > >
> > > **We have performed an additional experiment on CUB** with a pretrained feature extractor (ResNet18). With 250 prototypes, we achieve 89.1% accuracy (vs 91.7% for the CBM), showing again PGCMs come with only a small accuracy drop.
> > >
> > > &nbsp;
> > >
> > > > I appreciate the new experiment with 120 prototypes on CelebA. I had a followup question that how can a user set the number of prototypes m, if there is a systematic method apart from trial and error?
> > >
> > >
> > > Existing prototype works generally consider it a hyperparameter that is tuned in a trial-and-error fashion. When many prototypes are used, and the model in practice needs less capacity, this can easily be automatically detected, and prototypes can be pruned.
> > >
> > > In line with standard assumptions in interpretable machine learning, we treat certain design choices as *user-specified constraints reflecting human cognitive limits*. In our setting, the number of prototypes (m) is not purely a technical hyperparameter to be optimized, but rather an input that encodes how many distinct visual patterns a human can reasonably inspect and understand. This follows the **common paradigm in interpretable models where one first fixes what is considered human-understandable** (e.g. number of rules, concepts, or features), **and then designs the model to operate within these constraints**. From this perspective, selecting (m) is similar to choosing the size of an interpretable rule set or the number of concepts in a CBM: it is inherently user-dependent.
> > >
> > > In general, increasing (m) improves model expressiveness and can lead to better performance, but at the cost of reduced interpretability due to a larger number of prototypes to inspect. Therefore, **we recommend setting (m) as high as possible while remaining within the user’s capacity to interpret the resulting prototypes**.
> > >
> > > &nbsp;
> > >
> > > > Regarding the interpretability of the prototypes, the paper claims that PGCMs make the concepts more interpretable to humans. If a person tries to use the concept alignment table to intervene on the concepts then it is the user's ability and not the model design, even a small form of human evaluation or comparison between the prototypes and the CBM concept activations might help.
> > >
> > >
> > > We would like to clarify that **PGCMs *strictly extend* the functionality of standard CBMs.**
> > > In particular, **concept-level interventions in PGCMs are performed in exactly the same way as in CBMs (by modifying concept activations)**. Beyond this, PGCMs introduce an **additional level of intervention that is not available in CBMs**: users can directly inspect and modify the concept alignment table, i.e. the mapping between prototypes and concepts. This is a structural property of the model and does not depend on user skill: it provides strictly more accessible information about the model’s internal semantics. In this sense, **PGCMs' explanations form a superset of CBMs' explanations**, as they expose both the same concept activations as CBMs and additional grounded visual evidence. This additional structure enables a new form of semantic verification that is impossible in CBMs, where concept predictions remain opaque.
> > >
> > > To make this distinction clearer, **as asked by the reviewer, we will include a visualization** comparing CBMs and PGCMs on 3 examples (**figure**: https://figshare.com/s/23524aa8e58689667f92). We only show concepts predicted to be *present*.

---

### Official Review · Reviewer_uqfG · 2026-03-13

**Soundness:** 2
**Presentation:** 2
**Significance:** 2
**Originality:** 2
**Overall Recommendation:** 4
**Confidence:** 4

**Summary:**

Under the framework of concept bottleneck model, this work aims to align the learned concepts with the corresponding visual prototypes. The authors first extract image parts using a segmentation model. Then a prototype selector assigns a most similar prototype to this image part. The concept decoder maps the selected prototype to high-level concepts and a classifier predicts the image class based on the concepts. Experimental results are shown in concept prediction, image classification and concept interventions.

**Compliance With Llm Reviewing Policy:**

Affirmed.

**Final Justification:**

I think the response provided is a bit weak but is still acceptable to me. So I would like to raise the score.

**Key Questions For Authors:**

Can authors justify the importance of the concept annotations in a broader view and justify this work's contribution compared to prior works that have already aligned the visual prototypes with textual concepts?

**Limitations:**

yes.

**Strengths And Weaknesses:**

Novelty: there is an existing work that already handles the problem between the alignment between visual prototypes and textual concepts [1]. This work is not cited or compared with to discuss any additional contribution in this work.

[1] Align2Concept: Language Guided Interpretable Image Recognition by Visual Prototype and Textual Concept Alignment, ACM MM 2024

Method: [line 214] the reliability on an additional segmentation model weakens the contribution. Moreover, the training and architecture details of this segmentation model are missing (the appendix provides very vague descriptions). If this segmentation model is actually a pretrained large foundation model, the contribution will be further weakened: directly using the multi-modal foundation model would likely achieve even better concept-visual part alignment and achieve better accuracy, while obtaining much better generalization without requiring additional concept annotations.

Experiment design: the authors do not compare with any prototype based method [2]. [2] and its series of follow-up works are clearly proper baselines given the title “prototype grounded…” of this paper (at least from the aspect of accuracy). Many recent variants of concept bottleneck models such as [3] are not discussed or compared with neither. The comparisons are limited to very few early versions of CBMs.

[2] Chen, Chaofan, et al. "This looks like that: deep learning for interpretable image recognition." Advances in neural information processing systems 32 (2019).

[3] Oikarinen, Tuomas, et al. "Label-free Concept Bottleneck Models." The Eleventh International Conference on Learning Representations.

Experimental results: from figure 4 and table 4, it’s hard to say the intervention of the proposed approach is much better or the accuracy is really maintained.

Importance: the concept prediction part requires the concept label annotations, which is a major flaw of the conventional concept bottleneck models. The proposed framework is again constrained to the availability of such concept annotations, which makes the importance of the work weakened.

Discussions on [line 288-294] are also confusing. Most part-prototype networks learn explicit prototype representations [2].

---

> ### Author Rebuttal · Authors · 2026-03-30
>
> > From figure 4 and table 4, it’s hard to say the intervention of the proposed approach is much better or the accuracy is really maintained.
>
> **On ColorMNIST+, our method consistently achieves higher accuracy than all competitors for n>0 interventions.**
>
> **If the reviewer refers to accuracy in general: we conducted an additional experiment on CelebA with more prototypes.** This yields 78.6% concept accuracy, which is close to the CBM baseline (81.3%), showing that accuracy can be more closely maintained when sufficient capacity is provided. On ColorMNIST+, we already match the concept accuracy of CBMs.
>
> &nbsp;
>
> > Discussions on [line 288-294] are also confusing. Most part-prototype networks learn explicit prototype representations [2].
>
> > There is also an existing work that already handles alignment between visual prototypes and textual concepts [1]. This work is not cited or compared with to discuss any additional contribution in this work.
>
> **Existing part-prototype networks learn prototypes only in a latent space, not image-space prototypes.** The prototypes themselves are latent embeddings, and are only interpreted post-hoc by retrieving the most similar image regions according to the latent space. This provides no guarantees.
>
> **Our method learns and operates on image-space prototypes**, which are directly interpretable and can be inspected without relying on nearest-neighbor explanations. They are part of the forward pass of our model.
> This difference is akin to post-hoc explainability vs interpretability.
>
> [1] follows a ProtoPNet approach where prototypes are learned as latent embeddings and later explained by retrieving nearest training patches. **We will cite and explain this work in our Related Work section.**
>
> &nbsp;
>
> > The authors do not compare with any prototype based method [2], while they are clearly proper baselines given the paper title. Many recent CBM variants such as [3] are not discussed or compared with neither. The comparisons are limited to very few early versions of CBMs.
>
> **Such prototype-based methods** are not directly comparable to our setting, as they do not operate over a supervised concept space. They **cannot be used for the same purpose**.
>
> **Recent CBM variants** such as [3] **are completely orthogonal to our contribution, and thus do not really serve well as competitors**. For example, [3] focuses on how to obtain concept labels without human supervision, while other works improve the task predictor. In contrast, our work focuses on the concept predictor. These directions are complementary and can be combined. **We also note that we do compare against a recent CBM method** (CMR, NeurIPS 2024).
>
> We will discuss these other lines of CBM research in our Related Work.
>
> &nbsp;
>
> > Can authors justify the importance of the concept annotations in a broader view and justify this work's contribution compared to prior works that have already aligned the visual prototypes with textual concepts?
>
> In a broader context of CBMs, concept annotations are vital because they define a controllable, human-understandable semantic space. This is the case for all existing CBMs.
>
> Regarding prior works combining concepts and prototypes, we refer to our earlier answer: they do not allow verifying concept alignment.
>
> &nbsp;
>
> > The concept prediction requires concept labels, which is a major flaw of conventional CBMs.
>
> **PGCMs (and CBMs in general) are not constrained by human-annotated concept labels. Our framework is agnostic to the source of its supervision.** We can train our concept decoder using automated labels generated by Vision-Language Models like CLIP [1] without changes to our architecture.
>
> [1] Label-Free Concept Bottleneck Models.
>
> &nbsp;
>
> > The reliability on an additional segmentation model weakens the contribution. Moreover, the training and architecture details of this segmenter are missing. If this segmenter is actually a pretrained large foundation model, the contribution will be further weakened: directly using the foundation model would likely achieve even better concept alignment and accuracy without requiring concept annotations.
>
> **The segmentation model in our experiments is trained from scratch**. It is not a foundation model, but a standard U-Net with a ResNet backbone, trained using the ground-truth masks provided in the datasets. **We will expand the architectural and training details in the appendix to make this clear.**
>
> As noted earlier, PGCM can easily utilize VLM-generated labels if manual concept annotations are unavailable.
>
> **Foundation models are black boxes.** Directly using them to predict the task cannot achieve better concept alignment than PGCMs. **It is impossible to verify whether their internal predictions align with human semantics.** PGCM’s concept alignment table solves this exact problem.

---

> > ### Author Rebuttal · Reviewer_uqfG · 2026-04-02
> >
> > Regarding the response to implicit/explicit prototypes: [line 165-171, left column] lists the embeddings of the prototypes denoted as e. Do authors consider these representations not as prototypes in the designed framework? Besides, prior part-prototype networks finally have a prototype projection step, which replaces the learnable prototype parameters directly with an exact embedding of an image in the inference process. In this sense, I assume prior prototype based works do have the prototypes of the same type as this work?
> >
> > Regarding the concept annotations: I still feel this is a major limitation.
> > Overall I think it's quite tricky to train a segmentation model using annotations to conduct an interpretability work. To me it's acceptable to leverage these annotations as auxiliary metrics to evaluate interpretability. But directly using these annotations to train a segmentation model basically fully equals the better interpretations to better segmentations. If that is the correct direction, the interpretability area would be rather meaningless compared to directly training a strong segmentation model.

---

> > > ### Author Response · Authors · 2026-04-07
> > >
> > > > Regarding the response to implicit/explicit prototypes: (...)
> > >
> > > The embeddings e form the latent representation of a prototype, which we directly predict from the prototype's image representation (see e.g line 186+, col. 2). Thus, **our image representations of the prototypes are directly used in inference, and the embeddings *must* contain their information, as they are directly predicted from them.**
> > >
> > > **For existing part-prototype networks (e.g. ProtoPNet), their projection step operates entirely in latent feature space, not in pixel space.** Specifically, each prototype is replaced by the nearest latent embedding from a training image’s *feature map* (importantly, *not* from an image patch). This ensures that the prototype corresponds to an existing feature representation, but does not establish a direct correspondence to a localized pixel-space patch. Thus, existing prototype networks only identify which location in the feature map the prototype originates from; not a precise region in the input image. Due to the non-invertibility and spatial mixing inherent in convolutional networks, a latent feature vector generally aggregates information from a broad and overlapping receptive field, rather than a well-defined image patch.
> > >
> > > For instance, **ProtoPNet itself acknowledges this** gap when describing prototype visualization (see their paragraph on prototype visualization): the pixel-space patch "corresponding to a prototype" is obtained by upsampling the activation map and selecting the region of highest activation. **This procedure is heuristic and post-hoc**: it identifies the region that best matches the prototype, which may not be exactly the region that generated it.
> > >
> > > **Therefore, although existing prototype networks' projection ensures that prototypes are grounded in training data _in feature space_, their mapping to _pixel space_ remains approximate and post-hoc, and is not part of the model’s inference mechanism**. As such, these methods do not provide a guarantee that a prototype corresponds to a unique or exact image patch in pixel space.
> > >
> > > &nbsp;
> > >
> > > > Regarding the concept annotations: (...)
> > >
> > > We are unsure whether the reviewer refers to concept annotations (to train a concept predictor) or mask annotations (to train a segmentation model).
> > >
> > > **Both concept labels and a segmenter are a necessary cost to pay for interpretability and alignment verification.** In a general ML setting, we agree that the more supervision, the less scalable an approach is. However, we are not in a general ML setting, but in an interpretability setting. We are willing to spend 'more effort' to achieve what we care about: interpretability and alignment. In light of this, supervision must be seen not just as a way to train the model, but also to _align_ it to the human.
> > > - Concept labels are required to tell the model what semantics the latent neurons should have. This is the case for all existing CBMs.
> > > - A segmenter is necessary for our prototype learning, specifically to learn _image representations_ of prototypes (which existing prototype networks *don't* do). Learning prototype images is necessary for verifying and controlling alignment. Consider the famous Husky problem [1]: alignment is impossible without localization.
> > >
> > > Alignment can either be done to the human or to reference models. **If the human considers alignment to some reference models sufficient, then neither concept annotations nor the segmentation limit applicability.** In that case:
> > > - Concept annotations can be acquired through vision-language models, which means this requirement does not limit applicability at all [2].
> > > - Mask annotations are *not* necessary, and instead pretrained segmenters can be used, which are readily available.
> > >
> > > We really believe that good segmentation models are at a core component of interpretable models. However, they are _not_ interpretable models. **A strong segmentation model alone *cannot* replace interpretability research.** It does *not* have concepts, *nor* prototypes. Thus, it is *not interpretable* and cannot allow alignment verification. **Our method combines all these components to achieve these desiderata: segmentation and prototype learning (provide alignment verification), and concept learning (provides interpretability and control).**
> > >
> > > [1] “Why Should I Trust You?” Explaining the Predictions of Any Classifier.
> > >
> > > [2] Label-free Concept Bottleneck Models.

---

### Official Review · Reviewer_nqSQ · 2026-03-13

**Soundness:** 3
**Presentation:** 2
**Significance:** 3
**Originality:** 3
**Overall Recommendation:** 4
**Confidence:** 4

**Summary:**

This paper addresses a key problem in current CBMs: the false sense of interpretability. Standard CBMs assume that assigning a readable text label to a hidden concept node means the model actually understands the meaning. However, this is unverified and lacks visual proof. To solve this, the authors propose Prototype-Grounded Concept Models (PGCMs). PGCMs connect abstract concepts to actual visual image parts (prototypes) using a Concept Alignment Table. This makes the concepts visually verifiable and allows humans to directly edit the model's prototypes.

The motivation of this paper is great, and the prototype editing idea is highly promising. However, the paper overstates its performance, secretly relies on perfect segmentation masks, and uses a manual training trick that breaks its mathematical design. I am leaning towards a Weak Reject, but I am very willing to raise my score to an Accept if the authors honestly discuss the accuracy drop and provide baselines without perfect masks during the rebuttal stage.

**Compliance With Llm Reviewing Policy:**

Affirmed.

**Final Justification:**

The authors have demonstrated great academic integrity by acknowledging the limitations and trade-offs of their work while proving the distinct advantages of their prototype-level intervention paradigm. Because all my deal-breakers have been adequately addressed, I am happily raising my score to  Weak Accept.

**Key Questions For Authors:**

See Weaknesses.

**Limitations:**

No. Please explicitly add a short limitation paragraph to acknowledge:

1. The performance drop (accuracy-interpretability trade-off) compared to standard CBMs.
2. The heavy reliance on perfect ground-truth segmentation masks for complex datasets.
3. The risk of negative side effects when correcting multiple concepts at once.

**Strengths And Weaknesses:**

1. Strengths

S1. Strong Motivation and Insight: The paper brilliantly tackles the problem of linking concepts to real visual evidence. Pointing out that concept alignment must be visually proven, rather than just assuming it from high accuracy, is a very important insight for CBM research.

S2. Novel Intervention Method: Moving from changing concept numbers to "prototype-level editing" is highly innovative. The experiment on the noisy ColorMNIST+ dataset (Table 3) is very strong. Showing that humans can manually remove or edit bad visual prototypes to recover concept accuracy from 32.9% to 96.9% is an impressive result.

2. Major Weaknesses (Deal-breakers to be addressed in the Rebuttal)
W1. Overstated Performance Claims: In the Abstract (Line 27), the authors claim PGCMs "match the predictive performance of state-of-the-art CBMs." This is not true based on Table 4. On the harder CelebA dataset, PGCM's concept accuracy drops by 6.3% (from 81.3% to 75.0%) compared to the standard CBM, and task accuracy drops by 2.4%. Using the word "match" hides this performance drop and is misleading.
Actionable request: Remove the claim of "matching" performance from the Abstract and Introduction. Please honestly discuss this as an Accuracy-Interpretability Trade-off. Restricting the model to visual parts naturally limits its power. Discussing this 6.3% drop openly as the "cost of being verifiable" will make the paper much more trustworthy.

W2. Unrealistic Need for Perfect Masks (Critical Flaw): PGCM assumes we can easily extract meaningful local image parts. However, Appendix B (Lines 598-601) shows that for CelebA, the model unfairly uses perfect ground-truth (GT) segmentation masks provided by the dataset to crop out hair, noses, etc. In the real world, perfect masks are rarely available.
Actionable request: You MUST provide an experiment on CelebA without using GT masks. What happens if you use a standard, unguided segmentation model (like SAM or simple grid patches)? If the model extracts noisy or meaningless patches, does the Concept Alignment Table break down? Showing how the model works under normal, imperfect segmentation is required for acceptance.

W3. Mismatch Between Math and Actual Training: Sections 3.1 and 3.2 present a beautiful mathematical model (PGM) and loss function (Eq. 2), suggesting smooth, end-to-end learning. However, Appendix A.2 (Lines 580-589) admits that halfway through training, the learned prototypes are manually replaced with the nearest real training images and then frozen. This hard-coded step breaks the math presented in Section 3.
Actionable request: Move the "Prototype Swapping" step to the main text (Sec 3.3) and explain why it is needed (e.g., do math-only prototypes look too blurry?). Also, provide a "No-Swap" experiment to show what the prototypes look like if you only use the math from Section 3.4.

W4. Contradictory Evidence for "Simultaneous Concept Correction": The paper claims (Lines 76-78) that fixing one concept can automatically fix other concepts sharing the same prototype. But Figure 4 (CelebA) shows the exact opposite: the dependent intervention (PGCM*, dashed line) actually performs worse than standard independent intervention (PGCM, solid line).
Actionable request: Openly discuss the risk of "negative side effects." For facial features, concepts are highly connected. Fixing a prototype for "blond hair" might accidentally ruin a correct "smiling" prediction on that same prototype. Acknowledge this risk in complex images.

3. Minor Weaknesses (Presentation & Clarity)

M1. Missing Baseline Comparison in Intro: The Intro jumps directly to "learned visual prototypes" (Line 51). Why not just use standard heatmaps (like Grad-CAM)? Adding a brief sentence to explain that heatmaps cannot be easily edited by humans would make prototypes the clear best choice.

M2. Figure 4 Typos and Missing Takeaway: In the legend of Figure 4 (right, CelebA), there is a curve labeled PGCM+ that is never defined in the text (the text calls it PGCM* on Line 432). Also, the caption does not explain why the curves cross or what the reader should learn from it.

M3. Human Effort Limits: The method relies on humans manually checking the Concept Alignment Table. This is easy for 80 prototypes (CelebA), but the authors should briefly mention the limits of this manual check if a dataset has thousands of concepts and prototypes.

M4. Sudden Topic Change in Sec 3.3: The "Interpretability optimizations" part (Lines 207-217) appears suddenly. Please add a starting sentence explaining why pure math optimization isn't enough before explaining how you restrict it.

---

> ### Author Rebuttal · Authors · 2026-03-30
>
> We are happy the reviewer considers our motivation great, our idea highly promising and innovative with a strong motivation and insight, our mathematical formulations of our model beautiful, and brilliantly tackles the problem.
>
> &nbsp;
>
> > (Major W1) You claim PGCMs match the predictive performance of CBMs. This is not true. On the CelebA dataset, PGCM's concept accuracy drops compared to the standard CBM. Using the word "match" hides this and is misleading. Actionable request: Remove the claim of "matching" performance. Restricting the model to visual parts naturally limits its power.
>
> **We will change the claim** to make clear there is a small drop in accuracy.
>
> As with all prototype networks, the model's capacity is limited by the number of prototypes. By increasing it, the model will perform better (but less interpretable). **We have conducted an extra experiment with more prototypes for CelebA (120); PGCM then achieves performance closer to standard CBMs** (78.6% concept acc.).
>
> &nbsp;
>
> > (Major W2) The model unfairly uses ground-truth (GT) segmentation masks provided by the dataset. In the real world, perfect masks are rarely available. Actionable request: You MUST provide an experiment on CelebA without using GT masks. What happens if you use an unguided segmentation model? If the model extracts noisy patches, does the Concept Alignment Table break down?
>
> **In our work, segmentation should be viewed similarly to concept supervision: it is an increased effort, but is fundamental for alignment, which is our main concern.** We only use GT masks at training time, not at test time.
>
> We agree that perfect masks are not always available. A new experiment on CelebA was not feasible in the timeframe of the rebuttal. Instead, **we conducted an extra experiment on the CLEVR-Hans3 dataset using the pretrained SAM segmenter.** In this setting, SAM effectively works as a proxy for human alignment. The model achieves strong performance (99.2% concept acc.).
>
> Poor segmentation quality can harm alignment, but the key advantage of our approach is that **such issues can be detected through the alignment table, which is not possible in existing CBMs.**
>
> &nbsp;
>
> > (Major W3) Section 3 presents a beautiful mathematical model (PGM) and loss function. However, Appendix A.2 admits that halfway through training, the learned prototypes are replaced with the nearest training images. This step breaks the math of Section 3. Actionable request: Move this step to the main text and explain why it is needed, and provide a "No-Swap" experiment to show non-swapped prototypes.
>
> **The prototype swapping step is already mentioned in the main text** (line 193, col. 2), and we will move part of the appendix's explanation to the main text.
> In practice, prototypes without swapping already appear reasonable, but this is not guaranteed. **Swapping provides a guarantee that each prototype is understandable.** We explain this in line 196 (col. 2).
> **We will include the non-swapped prototypes in the appendix** using similar figures as https://figshare.com/s/ef21721011621d8027e7.
>
> &nbsp;
>
> > (Major W4) The paper claims that fixing one concept can automatically fix others sharing the same prototype. But Figure 4 (CelebA) shows the opposite: the dependent intervention actually performs worse. Actionable request: Openly discuss this risk.
>
> **This is not unique to our approach, but is a risk with all CBMs that model concepts jointly** rather than independently. **We will acknowledge this risk in the conclusion as:** "A possible risk that PGCMs share with other CBMs with joint concepts is that a single intervention may wrongly change other concept predictions".
>
> &nbsp;
>
> > (Minor M1) The Intro jumps directly to "learned visual prototypes". Why not just use heatmaps? Adding a brief sentence to explain that heatmaps cannot be edited by humans would make prototypes the clear best choice.
>
> **We will add:** "Standard heatmaps only provide post-hoc explainability as opposed to interpretability, and e.g. do not allow model interventions, making prototypes the clear choice".
>
> &nbsp;
>
> > (Minor M2) In the legend of Figure 4, there is a curve labeled PGCM+ that is never defined in the text (the text calls it PGCM*). Also, the caption does not explain what the reader should learn from it.
>
> **We believe the legend does correctly refer to PGCM\*.**
>
> We will add to the caption: "On ColorMNIST+, PGCM* shows higher intervenability than competitors; on CelebA, PGCM's performance eventually catches up with the competitors."
>
> &nbsp;
>
> > (Minor M3) Humans must manually check the Concept Alignment Table. This is easy for 80 prototypes, but the authors should briefly mention the limits of this manual check if a dataset has thousands of concepts and prototypes.
>
> **We agree and will acknowledge this limitation in the conclusion. However, we want to stress that this limitation is shared by all CBMs**: CBMs with very large numbers of concepts are inherently difficult to interpret.

---

> > ### Author Rebuttal · Reviewer_nqSQ · 2026-04-03
> >
> > The authors has largrly adressed my concerns.

---

### Decision · Program_Chairs · 2026-04-30

**Decision:**

Accept (regular)

**Comment:**

While the overall balance of reviewer ratings on this paper is leaning positive (three weak accept, one weak reject), the concerns expressed in reviews seem significant, and should not be considered resolved through a potentially rushed rebuttal discussion, and based on potentially rushed multiple critical results that were not in the original submission. For example, multiple reviewers are concerned that the only result in the original paper that is not on a toy dataset, shows a significant drop in performance. Further, there is no quantitative result (other than on ColorMNIST+), that shows the proposed work outperforms prior ones. However, prior work (e.g. CMR by Debot et al.) seems to also aim to show that their proposed approach does not lead to a big drop in performance. Beyond this critical issue, there are others, e.g. about comparison to several prior works, and the lack of a user study (which is also missing in the cited prior work).